# Digger wasps *Microbembex monodonta* SAY (Hymenoptera, Crabronidae) rely exclusively on visual cues when pinpointing their nest entrances

**Matthew J. Cormons[1], Jochen Zeil[2]***

**1** 26201 Dennis Dr, Parksley, Virginia, United States of America, **2** Research School of Biology, The Australian National University, Canberra, ACT, Australia

* Jochen.zeil@anu.edu.au

**Data Availability Statement:** All relevant data are within the paper and its Supporting Information files.

## Abstract

The ability of insects to navigate and home is crucial to fundamental tasks, such as pollination, parental care, procuring food, and finding mates. Despite recent advances in our understanding of visual homing in insects, it remains unclear exactly how ground-nesting Hymenoptera are able to precisely locate their often inconspicuous or hidden reproductive burrow entrances. Here we show that the ground-nesting wasp *Microbembex monodonta* locates her hidden burrow entrance with the help of local landmarks, but only if their view of the wider panorama is not blocked. Moreover, the wasps are able to pinpoint the burrow location to within a few centimeters when potential olfactory, tactile and auditory cues are locally masked. We conclude that *M. monodonta* locate their hidden burrows relying exclusively on local visual cues in the context of the wider panorama. We discuss these results in the light of the older and more recent literature on nest recognition and homing in insects.

## Introduction

The ability of nesting insects to reliably and efficiently pinpoint their nest entrances when returning from foraging excursions has been and continues to be the focus of much research [1–5]. For digger wasps, Tinbergen [6, 7] had established the importance of vision within the burrow vicinity as the modality used by the insects to locate their reproductive burrows. Since then, it has been learned that homing Hymenoptera make use of the entire panorama, which includes distant cues, as well as cues within the nest vicinity (local cues) [8, 9], and that these cues are used simultaneously in order to home precisely to the burrow [10]. The importance of vision in homing and in pinpointing nests have been reported for other Hymenoptera, such as ants and bees [11–38].

Various digger wasps species use naturally-occurring objects within the burrow vicinity (local cues) to precisely locate their burrows. The shifting of such local objects causes homing wasps to dig for their burrows relative to the shift [e.g. Marchand in 12 (p125), 14, 15, 39, 40]. In addition, introduced objects, both natural (e.g., pine cones) and man-made (e.g.,wooden

**Funding:** The author received no specific funding for this work.

**Competing interests:** The authors have declared that no competing interests exist.

cubes and flat rings), placed in the burrow vicinities of various digger wasp species, can be learned as cues for locating nest entrances. As with naturally-occurring local cues, wasps followed introduced cues when they were shifted and searched for their burrows relative to the shifted objects [6, 7, 10, 17, 20, 22, 26, 40 in 41]. Cartwright and Collett [25, 42] have shown that this search behaviour can be explained by assuming that insects memorize the view from the nest entrance (as if taking a snapshot) and upon returning to the nest area moving in such a way as to minimize the mismatch between what they currently see and their memorized snapshot. This snapshot model of homing can be generalized to global image matching without identification of individual objects in a panoramic view from the nest [3, 5, 43]. The crucial insight here was that a location in a natural environment is uniquely defined by the view taken from it because global image differences–in the simplest way measured as the root mean squared pixel difference between a memorized goal image and the images seen on approach to the goal–become systematically smaller as the distance to the goal decreases. Pinpointing the nest entrance, thus becomes a gradient descent in global image differences [43, 44, see also 45,49,50,51] and experiments with large screens occluding part of the scene have added to other evidence (see examples in [1]) that navigating insects do make use of the full visual panorama and not only of individual landmarks [9]. However, whether it is the increasing salience of local visual cues that guide the final approach to the nest or whether pinpointing the nest location requires an 'attentional' switch to fine visual details, such as the nest entrance itself or visual features around it, is not known at present.

Moreover, in most cases it remains also unclear to what extent other cues help ground-nesting insects in their final approach to the nest. Desert ants, for instance, are guided by the $CO_2$ plume emanating from the nest entrance [52] and other, very close-range cues such as local surface topography or even the sound of larvae emanating from the nest [e.g. 53] may help ground-nesting insects to pinpoint the exact location of the entrance. The role of such non-visual cues may be particularly important for insects such as the *Microbembex monodonta* we studied here, which return to a nest the entrance of which they had previously carefully closed, covered and camouflaged.

*Microbembex monodonta* is a solitary digger wasp that nests in aggregations of reproductive burrows (nests) in sandy areas with sparse vegetation. Each nest consists of an entrance to an oblique burrow that ends in a terminal cell. Upon completing her burrow the wasp will deposit a single egg in the cell, then kick sand over the burrow entrance, making it visually impossible to locate, certainly to a human investigator, as well as to potential predators and parasites [54]. She then leaves to collect provisions for her prospective larva. Returning to her burrow vicinity, she flies low, always facing forward (the direction of the burrow from the entrance) and without hesitation lands, digs precisely at the invisible burrow entrance, and enters [22, 55].

Here, we report on a series of experiments, asking whether *Microbembex monodonta* wasps locate their hidden burrow entrances relative to small local landmarks and whether the view to the wider panorama is needed for pinpointing the burrow. By studying the wasps' search when their nest entrances were covered in a variety of ways we also tested whether local visual, olfactory and auditory/vibrational cues help wasps to locate their hidden nests.

## Methods

The study area was a vegetated, sandy blowout covering an area estimated to be about 3,600 m² (see images in S1 Fig for visual appearance). At the time, it had been part of an abandoned farm located north of the town of Spring Green, Sauk Co., Wisconsin. The entire farm has since been established as the Spring Green Preserve (46.172763N, 89.579446W). All the experiments discussed in this paper were conducted in this area.

Subject wasps were netted at their burrows, then color-coded for positive identification using Testors® lacquer model paints. The following experiments were conducted to test for: 1) the influence of local visual cues, 2) the influence of distant cues, 3) the interaction between local and distant cues, and 4) the influence of non-visual cues.

## 1) The influence of local visual cues

A triangular configuration of three unpainted, tapered cork stoppers (3.5 cm high and 5.5 cm base diameter) was placed around the burrow entrances of five wasps while each was away foraging for provisions. Each wasp was allowed to return undisturbed, enter, and leave. This is the phase in which the wasp is trained to recognize the objects. The configuration was then shifted within a range of 3.4–6.75 cm from the burrow, in a random position (left, right or forward).

## 2) The influence of distant visual cues

In this experiment an opaque visual barrier was placed behind the nests of six wasps, which blocked large parts of the distant panorama as seen from the approach direction of the returning insects (see S2 Fig); no local visual cues were disturbed or introduced. The barriers were at least 60 cm high, but differed in configuration for each of the wasps and could be moved sideways to cover or uncover different parts of the panorama.

## 3) The interaction between local and distant cues (the barrier-cube configuration experiment)

The following experiment was conducted to check whether *M. monodonta* were able to use cues within their burrow vicinities to precisely locate their hidden burrows in the absence of the more distant visual background. The experimental apparatus consisted of an opaque barrier (100 x 60 cm), two side pieces (48 cm$^2$) of the same material (see S2 Fig) and three wooden cubes (6 cm$^3$) painted with Rustoleum® flat black. Each of 10 wasps was tested individually as follows.

Each wasp was permitted to enter her burrow with provisions, undisturbed. She was netted, either as she began digging at the burrow entrance, or upon leaving, then marked, and released. While she was away foraging, a triangular configuration of three cubes was placed around her burrow entrance. Small plants and debris within approximately 50 cm surrounding her burrow were removed in order to eliminate the possibility of the wasp using any obvious naturally-occurring visual features to pinpoint the nest, in lieu of the introduced cues. The wasp was allowed to get used to this disturbance and to return, enter, exit, and cover her burrow. To ascertain she had learned to use the cube-configuration as a homing cue the latter was shifted 6 cm to the right while she was again away foraging, and the area covered with a thin layer of sand. Her response when she returned was recorded. While she was again away, the opaque barrier was positioned 50 cm behind the actual burrow entrance and the cube-configuration shifted 12 cm left of its previous position. Her response on her return was recorded. The barrier was then removed (without shifting the configuration) and her response on her return noted. This procedure was followed with each of the 10 wasps.

## 4) The influence of non-visual cues

**Experiment 1.** This experiment was designed to mask or alter local homing cues within the burrow vicinity—visual, olfactory, tactile, and possibly auditory/vibratory (Fig 1). The distribution of repeated digging attempts of returning wasps was recorded with the expectation that the absence of crucial cues would lead to no or more widely distributed digging attempts. Seven different covers were placed over the burrows of 12 wasps, each designed to mask or

| CONDITION | VISUAL (SAND RELIEF) | VISUAL (OTHER) | OLFACTORY | AUDITORY |
|---|---|---|---|---|
| CONTROL | O | O | O | O |
| A: plexiglass-paper-cloth-sand | X | X | X | X |
| B: plexiglass-paper-cloth-frame | X | X | X | X |
| C: paper(window)-wax paper-cloth-frame | X | X | X | X |
| D: paper (hole)-cloth-frame | X | X | O | O |
| E: cloth alone hugging surface | O | X | O | O |
| F: plastic-cloth hugging surface | O | X | X | X |
| G: cloth over frame | X | X | O | O |

**Fig 1. Table 1.** Cues affected by different layers of materials covering the nest entrance. Layers of different materials as indicated under CONDITION were placed over the nest entrances of wasps which left visual, olfactory or auditory cues masked (X) or unmasked (O).

alter at least one potential homing cue that might help the wasps to precisely locate their burrow entrances. Each cover is referred to as condition and the name of each condition begins with the part closest to the land surface, followed in upward placement order of the subsequent parts. The conditions are shown in Fig 2.

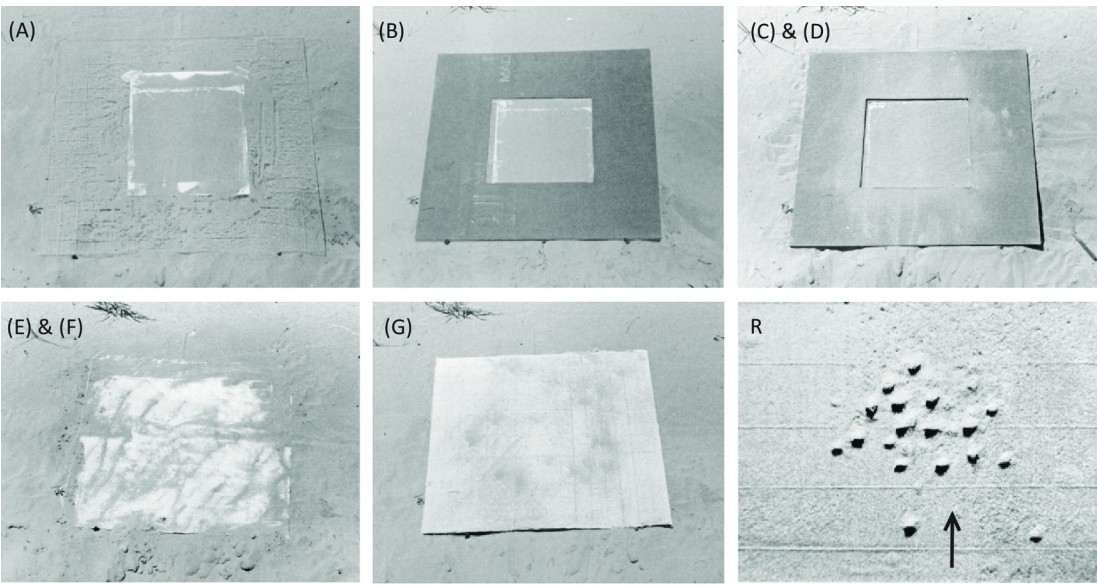

**Fig 2. Investigating local burrow location cues I.** Twelve returning wasps were confronted with seven different covers placed over their burrows. A—G refer to the experimental conditions (see Methods for details and Fig 6 for results). **Condition (A):** Three layers consisting of a plexiglass board, paper, and cloth to exclude olfactory, auditory and tactile cues. Overall apparatus height is 0.3 cm. **Condition (B):** Same as A, but with an additional frame excluding subtle changes in the sand-surface relief. Overall apparatus height is 0.6 cm. **Conditions (C) & (D):** As in B, but without plexiglass board. Overall height of each is 0.3 cm. **Condition (E):** With single cloth layer, and **Condition (F):** Flexible plastic sheet under cloth. Apparatus height for both is 0 cm. **Condition (G):** Cloth stretched over frame. Height is 0.3 cm. **R** Shows the distinct traces left by repeated digging attempts of a returning wasp, using only distant cues, searching for her hidden burrow entrance (grid shows 3.5 cm$^2$ squares).

Condition (A)—*plexiglass-paper-cloth-sand*. The apparatus consisted of a plexiglass board, 48cm$^2$ x 0.3 cm, inscribed with a grid of 16x16 squares with 3.0 cm side-length, covered by a sheet of white typing paper measuring 21.6 cm x 27.9 cm, printed with a grid of 7x7, 3.0 cm squares; a 21 cm$^2$ unmarked linen cloth placed over the latter; and sand from the perimeter of the nest vicinity sprinkled over the linen cloth. This condition, in particular due to the plexiglass board, masked the sand relief, other visual, olfactory and auditory/vibrational cues.

Condition (B)—*plexiglass-paper-cloth-frame*. Similar to Condition (A), but with the addition of a 48 cm$^2$ x 0.6 cm fiberboard frame with 22.4 cm square cut from its center, and a new sheet of 21.6×27.9 cm typing paper printed with a grid of 7×7, 3.0 cm squares. The total apparatus height measured 0.6 cm, twice as high as condition (A). This condition, again because of the plexiglass board, also masked the sand relief, other visual, olfactory and auditory/vibrational cues.

Condition (C)—*paper (window)-wax paper-cloth-frame*. This is much like condition (B), differing as follows: 1) no plexiglass board (so the apparatus was lower (0.3 cm); 2) a 0.75 cm diameter hole punched in the paper; 3) a sheet of waxed paper (Cut-rite®) stapled to the back of the grid-sheet creating a sealed "window" over the hole. All was covered by the cloth and sand as in the previous conditions; the level of the paper lay below the level of the frame. This condition masked visual other than relief and olfactory cues, but also the sand relief and auditory/vibrational cues.

Condition (D)—*paper (hole)- cloth-frame*. This condition differed from condition (C) in that the window is now an open hole. This would enable any olfactory and auditory/vibrational cues to emanate, unlike the sealed window in condition (C)). This condition thus masked other visual and sand relief cues, but left olfactory and auditory/vibrational cues unmodified.

Condition (E)—*cloth alone*. The cloth covered the burrow vicinity as in all previous conditions. A grid of 16x16 squares of 3 cm side-length was drawn on the cloth with a graphite pencil, and sand from the nest vicinity sprinkled over the cloth so that the grid lines were still sufficiently visible to the experimenter. In this and the following condition (F), the cover hugged the sand surface relief. This condition masked visual other than relief cues, but left relief, olfactory and auditory/vibrational cues unmodified, although we cannot be certain about the extent to which olfactory and auditory/vibrational cues may have been attenuated by the cloth.

Condition (F)—*plastic-cloth*. This differed from condition (F) by the addition of a thin flexible, impermeable, plastic sheet beneath the cloth. This condition left relief cues unmodified but the the cloth masked other visual and the plastic sheet masked olfactory and auditory/vibrational cues.

Condition (G)—*cloth over frame*. The cloth used in conditions (F) and (G) was tightly stretched over the frame used in conditions (B) to (E) and placed over the burrow vicinity. This condition masked relief and visual other than relief cues, but the cloth left olfactory and auditory/vibrational cues unmodified, although we cannot be certain about the extent to which olfactory and auditory/vibrational cues may have been attenuated by the cloth.

Each subject was chosen by observing her land and successfully dig at her undisturbed burrow entrance. While the wasp was in her burrow, in order for the experimenter to locate the burrow entrance (which she covers upon leaving), two thin lines crossing at 90˚ extending beyond the cover and intersecting at the burrow entrance were lightly inscribed to mark where the entrance was when covered. While the wasp was away again, one of the seven experimental conditions was randomly selected and placed over her burrow. Each time the wasp returned with provisions and attempted to dig the precise point was marked, either on the paper itself, or when the paper was not part of the apparatus [conditions (F), (G), and (H)], by marking a

paper grid on a hand-held clipboard. Fifteen digging attempts were recorded for each wasp tested (e.g. Fig 2R), then the apparatus was removed to allow each wasp to enter her burrow. When the wasp left again, a different condition was randomly chosen and the wasp's digging attempts recorded when she returned. This procedure was repeated until all seven conditions were presented to each of the 12 wasps. Fig 1 shows what cues were affected by each condition. It is important to note that information on wasp ID and on the distribution of individual digging attempts has unfortunately been lost. However, the data set is balanced, because each wasp contributes the same number of data (mean and spread of 15 digging attempts) to each condition and each of 12 wasps was confronted with each condition.

Statistics: Linear models were fit with condition as predictor variable separately to the x- and y-coordinates relative to the true burrow location of the means of the search distributions and to the width in x- and y-direction of the ellipses of equal probability covering 99% of the digging attempts of each wasp. Results were tested with ANOVA. Because graphical inspection of residuals showed deviations from normality in some cases, we performed in addition a permutation test, permuting condition 5000 times, using the F-statistics of the linear models as the permutation statistics. A paired t-test was used to test for differences between the x- and y-directions of means and spread across all conditions. Analysis was carried out in Matlab 2022b (Mathworks, Natick, MA, USA).

**Experiment 2.** The objective of this experiment was similar to the previous, but offered an air-tight seal that excluded olfactory and auditory cues, including changes in air-flow or sound reflections between solid sand and the loose sand over the hollow burrow. The apparatus consisted of a transparent, colorless. 0.6 cm thick plexiglass tray measuring 5.9 cm x 5.9 cm and was 3.4 cm high. The edges were sharpened to facilitate pushing the inverted tray into the sand (Fig 3). Fourteen wasps that were observed to successfully enter their burrows, exit, cover, and leave to forage were selected. While a wasp was within her burrow the latter was marked by imprinting a shallow circle around the entrance using a finger (shown in Fig 3). While the wasp was away foraging, the inverted tray was placed over the burrow, with the burrow entrance centered, and pushed into the sand; spurts of sand flying up from one or more sides indicated a virtually air-tight seal. To improve on the seal water was poured along the sides to soak into the sand (Fig 3).

## Results

Local landmarks influence where wasps search for their hidden burrow entrances (Fig 4A): Five wasps were allowed to become accustomed to finding their burrow in the center of an array of 3 cork stoppers ('training') and on subsequent returns were confronted with a shifted array of these landmarks. In each shift-trial wasps eventually located their burrow. All responded similarly to the one shown in Fig 4A. Before attempting to dig for their burrows the wasps reacted by flying around the shifted configuration of local landmarks for two to three minutes, almost always where the configuration led them and their initial digging attempt was always at a location where, relative to the landmark array, they had found their burrow on their previous return (see [56]) for more evidence of this rapid location learning). Eventually the wasps corrected themselves and dug at their burrow entrances, which had not been disturbed; they did so by first hovering over the place they had been misled to dig, then flew directly above the actual burrow entrance, landed, and dug.

To investigate the influence of distant visual cues six homing wasps were confronted with opaque barriers behind their hidden burrow entrances. All but one of the wasps never attempted to dig; they appeared to be lost. The single wasp that attempted to dig did so at both ends of the barrier, which were at least 50 cm from the burrow entrance (Fig 4B). When the

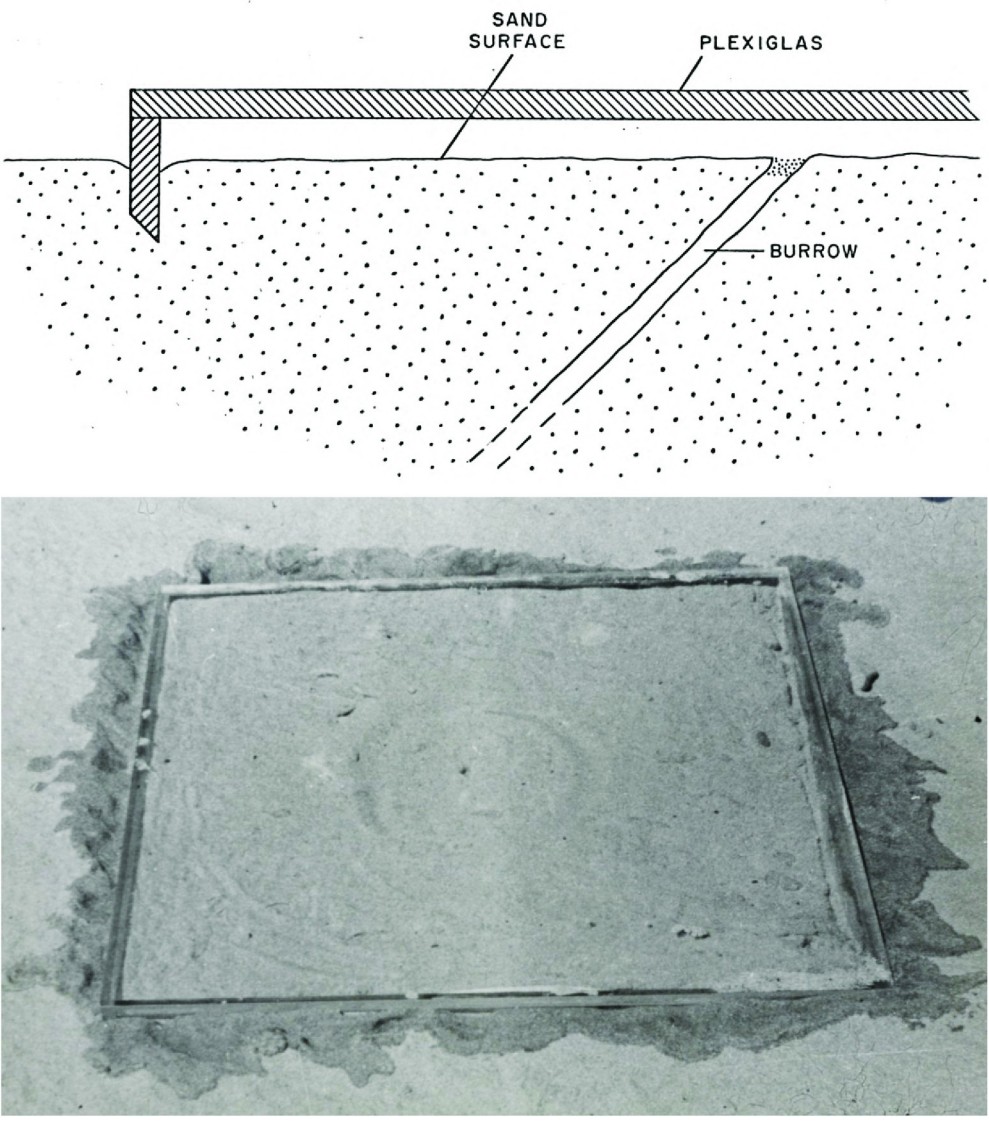

**Fig 3. Excluding all possible local cues to burrow location except local visual cues.** A large, clear plexiglass frame was placed over a burrow with vertical side-walls pushed into the sand without disturbing the local sand relief around the burrow. All returning wasps searched for the burrow entrance at the center of this frame.

barrier was moved to the left, so that the right side was closer to the burrow entrance the wasp dug at the right end of the barrier, and *vice versa* with a shift to the right (Fig 4C).

To investigate how local and distant cues interact when wasps attempt to locate their hidden burrow entrances, their response to small shifts of local landmarks was tested in the presence of a large opaque barrier. Both digging attempts at the center of a three-cube landmark array and hovering above it were considered as an indication that the wasps used the visual appearance of the landmarks to locate their burrow. The barrier set-up for these experiments is shown in Fig 5A and the sequence of tests with local visual cues in Fig 5B. Results for all 10 wasps are shown in Fig 5C. All 10 wasps learnt to use the cube-configuration as a homing cue after a single exposure (Fig 5B, a). When tested with the shifted cube-configuration (Fig 5B, b)

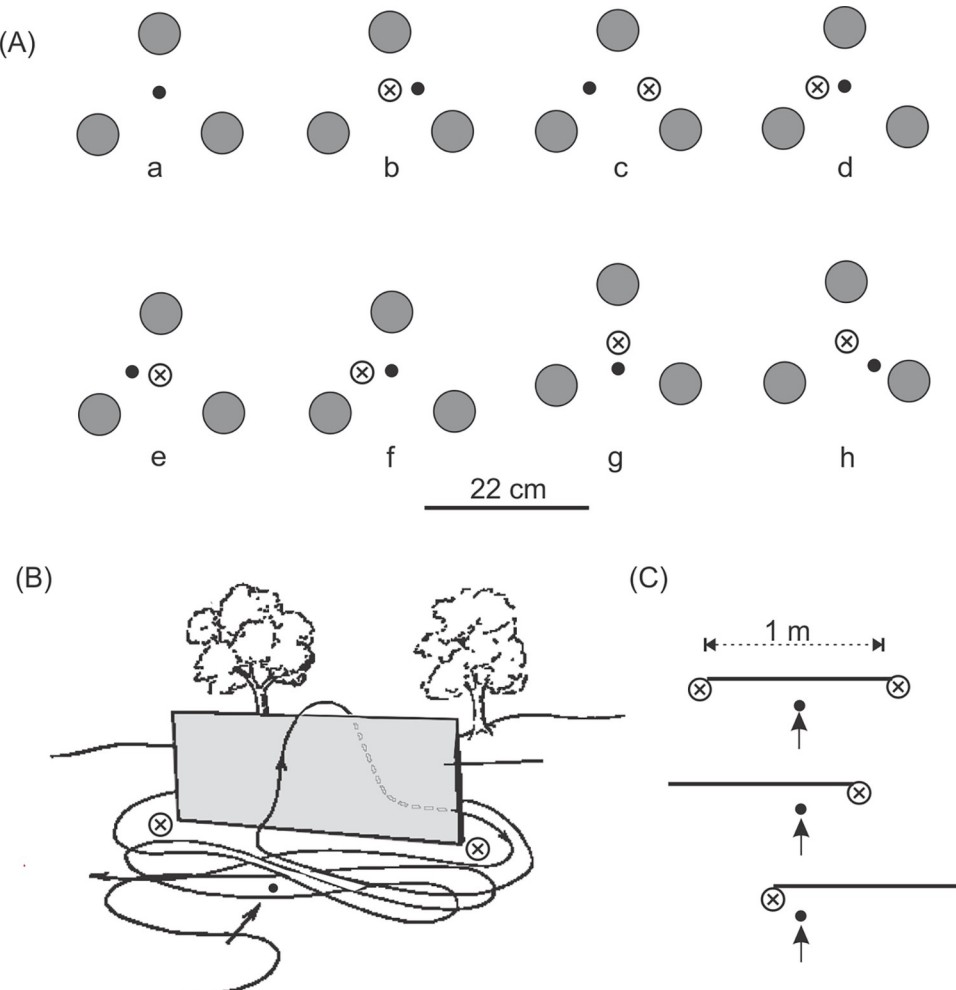

**Fig 4. The influence of local and distant cues.** (A) A typical example of how one of 5 wasps responded to shifts of local landmarks. Three cork stoppers (gray circles) were introduced around the burrow entrance (black dots) of a wasp. (a) training trial for the wasp to learn introduced cork-configuration cue surrounding the burrow prior to first shift. (b)–(h) consecutive configuration-shifts and digging attempts (⊗) of this particular wasp, which eventually does find her burrow; each initial digging following a subsequent shift was where the wasp was last successful, relative to the configuration. At (g) the wasp was not permitted to find her burrow before the next shift, resulting in her digging relative to configuration in (h) at a location where she had found the burrow in (f). (B) A wasp responding to the presence of an opaque barrier behind the burrow, with digging attempts at either end of the barrier with unimpeded view of the wider surrounding. (C) Same wasp digging at the ends of the barrier when the latter is centered on the burrow, then digging at either end, when the barrier is shifted to the left and right of the burrow.

8 wasps first dug at the center of the shifted array before locating their burrow. The two that did not follow landed and dug without hesitation at their actual burrows. One of the eight wasps that located their burrow did not do so immediately; she hovered over the point where the cubes misled her. However, she quickly shifted to her actual burrow location, landed, and successfully dug.

Once the barrier was in place, masking frontal, and some lateral, distant cues all 10 wasps failed to dig at the center of the landmark array, none hovered above it and none did find her burrow (Fig 5C, c). The wasps flew back and forth in front of the barrier, around, and over it (see Fig 4B), and even flew among the cubes. They never attempted to dig, never hovered over any particular area, nor otherwise indicated any recognition of the cube-configuration; they

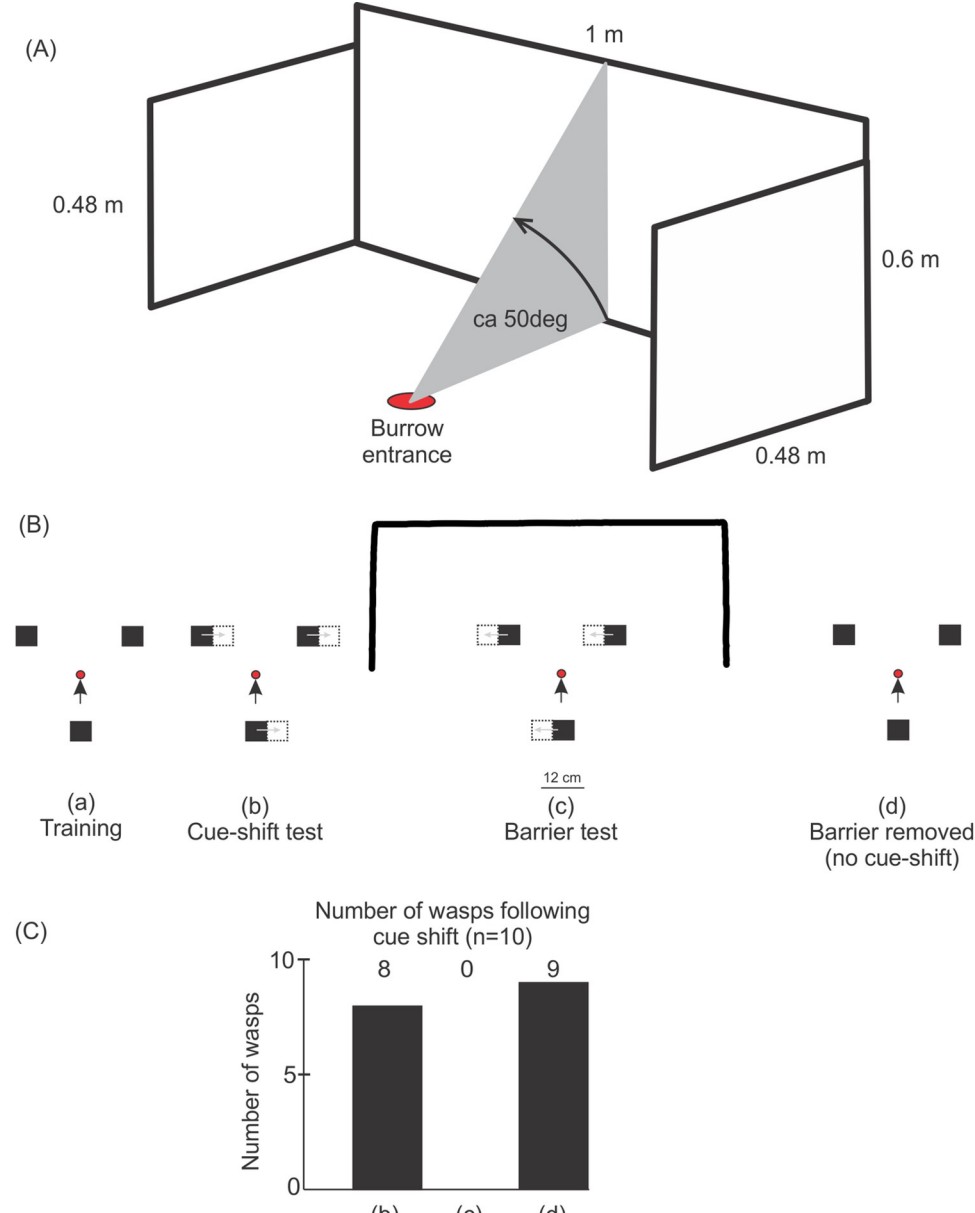

**Fig 5. The interaction between local and distant cues.** Ten wasps were trained on a configuration of three 3 cm³ wooden cubes placed around their burrows. **(A)** Position and measurements of the opaque barrier. **(B)** (a) training to configuration (black squares) around burrow, (b) sideways configuration-shift to the right (white squares) to test learning by wasp, (c) introduction of barrier along with shift of configuration to left of original configuration position, (d) barrier removed (configuration not shifted), black arrows indicate direction wasps face. **(C)** The number of wasps responding to the shift of the landmark configuration by digging attempts or by hovering at the center of the landmark array in condition (b), (c), and (d) (see (B) above). None of the wasps followed the configuration in (c).

appeared to be lost. Yet, all 10 wasps had been able to enter the vicinities of their burrows. When the barrier was removed nine of the 10 wasps pinpointed the nest in the center of the landmark array without difficulty; the tenth wasp did not return.

In the final set of experiments various combinations of materials were used to cover the hidden burrow entrances of wasps with the aim of excluding visual, olfactory and potential

auditory/vibrational cues associated with the sand-surface in the burrow vicinity and with the hidden burrow entrance itself. The ellipse graphs in Fig 6 show the distributions of 15 digging attempts of each of 12 wasps for each of the seven conditions described in the method section and in each of the panels in Fig 6. Each condition graph shows the areas for 99% of all attempted diggings as grey ellipses and the means of the search distributions as coloured dots. Greater homing accuracy is shown by smaller ellipses centered on the burrow entrance location. We compare the search distributions in Fig 6H. The top row of Fig 6H shows the distributions of the distances in x- and y-direction (as marked in Fig 6G) of the means of the search distributions from the true burrow location for all wasps across all conditions in the left panel and for each condition separately in the middle and right panel. The same is shown for the spread of the search distributions (the size of grey ellipses in x- and y-direction) in the bottom row of Fig 6H.

Overall, the wasps in all conditions were digging within a few centimeters of their true burrow location, but slightly short of the burrow in y-direction (Fig 6H, top left panel, x and y distributions are different: $p \ll 0.05$, paired t-test). The spread of their search distribution across all conditions was also different in x- and y-direction (Fig 6H, bottom left panel, $p \ll 0.05$, paired t-test). The distributions of the distances of means from the burrow location in both x- and y-direction did not differ across conditions (Fig 6H, top middle panel, x: ANOVA $F = 1.94$, $p = 0.08$; permutation $p = 0.17$; Fig 6H, top right panel, y: ANOVA $F = 1.52$, $p = 0.18$; permutation $p = 0.22$). Equally, spread across conditions was also not different (Fig 6H, bottom left panel, x: ANOVA $F = 0.25$ $p = 0.96$; permutation $p = 0.53$; Fig 6H right panel, y: ANOVA $F = 0.15$ $p = 0.99$; permutation $p = 0.69$). In brief, the distributions of wasp digging attempts did not differ depending on condition.

In the most radical attempt to exclude non-visual cues in the burrow vicinity, a plexiglass frame with vertical sidewalls was placed over the burrow entrances of 14 wasps (Fig 3). All 14 wasps returned and attempted to dig on the inverted tray apparatus, apparently directly over the burrow entrance (a homing wasp is seen as a dark spot in Fig 3). None of the wasps attempted to dig along the sides of the box.

## Discussion

The experiments presented here demonstrate that *M. monodonta* wasps very quickly learn to use introduced local visual cues to locate their burrows. When these cues are shifted the wasps' first digging attempts were always at the location they were last successful relative to the configuration. However, when the more distant visual panorama was blocked by barriers, wasps were unable to locate their burrows, even when the local cues they previously followed were still present. Finally, wasps were able to locate their hidden burrow position to within a few centimeters when all possible cues (other than distant visual cues) were excluded by different layers of materials covering the nest entrance, including an airtight cover. *M. monodonta*, like many other nesting insects, thus rely exclusively on local visual cues in the context of the wider visual panorama to precisely locate their hidden nest entrances.

These results confirm a large body of literature on the homing abilities of insects, but also raise a number of interesting questions regarding the accuracy of visual homing and the relationship between global visual cues and the fine visual details close to the goal. Lastly, we discuss the question why other, especially olfactory, cues do not play a role in nest entrance recognition.

### Visual homing

Inspired by Tinbergen's seminal experiments [6, 7], the literature is rich with studies investigating the abilities of insects to navigate between places that are significant to them (honeybees

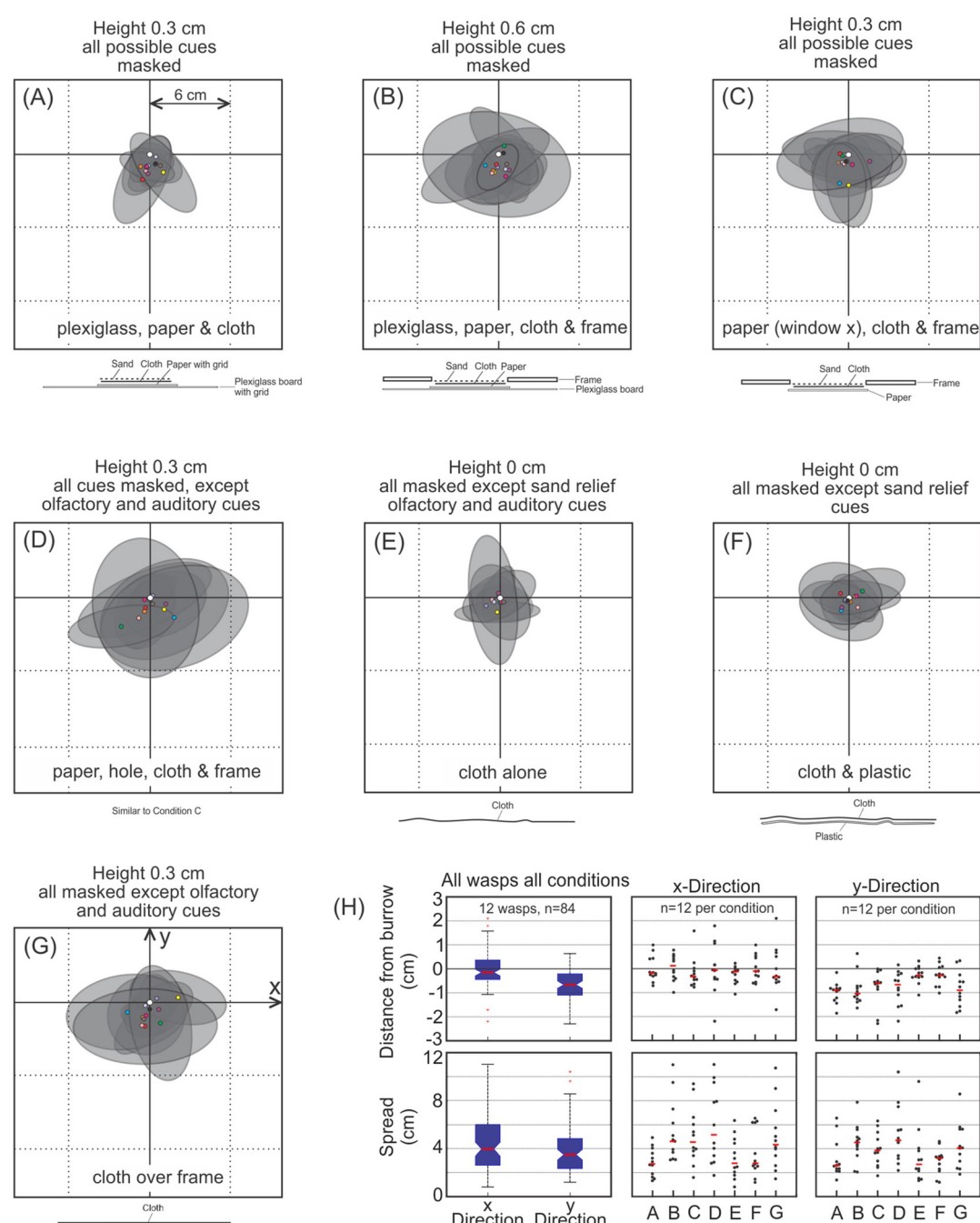

**Fig 6. Investigating local burrow location cues II.** The distribution of 15 digging attempts (see Fig 2R) of each of 12 wasps returning to their locally altered burrow environment. Grey ellipses are ellipses of equal probability covering 99% of the digging attempts of each wasp. Coloured dots mark the means of these distributions. Burrow entrance marked with a white circle at the intersection of x- and y-axes as labeled in G. (A)—(G) Show the distributions for the 7 experimental conditions. (H) Left column panels show boxplots of x- and y-distances (top) and spread in x- and y-direction (bottom) from true burrow location across all conditions. Medians are indicated by horizontal red lines, box size shows 25th and 75th percentiles, whiskers extend to the most extreme data points not considered outliers and outliers (points more than 1.5 times the interquartile range away from the top or bottom of the box) marked by red crosses. Middle and right column panels show for each condition separately the individual means in x- and y-direction (top row) and ellipse extent or spread in x- and y-direction (bottom row). Medians are marked by horizontal red lines. See text for statistical results.

[25, 42, 47], wasps [13–17, 21, 26, 28], ants [4, 5, 9, 24, 29, 30], flies [45, 46, 57], and crickets [58]; reviewed in [1, 3, 8, 32, 38, 48].

The first attempt in the insect literature to model this ability was Collett's and Cartwright's snapshot model [42] suggesting that insects move in such a way as to minimize the difference between a memorized snapshot and what they currently see. The elegance of the model was to also suggest a procedure how this could be done by a move and compare process that minimized first the azimuth mismatch between remembered and currently seen objects by changes in orientation and subsequently the apparent size mismatch by moving toward objects that appear smaller than remembered and away from objects that appear larger than remembered. It is important to note that the model assumed insects to be able to segment a scene into individual landmarks. This, however, may be difficult in visually complex natural environments and may actually not be needed, because a similar move and compare strategy operating on global image differences can serve the same purpose of pinpointing a location with the aid of remembered views [43, 59].

The accuracy with which panoramic views provide information on location depends on the distance of visual features [43, 60] and on the sensitivity of insects to image differences. Close to the burrow, the image differences experienced by homing insects will depend on the visual salience of nearby objects, in terms of their angular size and contrast with the background. Tinbergen, for instance, found that the effectiveness of homing cues depends on aspects such as size, height, contrast with surroundings, color, texture, edges, shapes, patterns, surface area and even shadows [20]. Some investigators found the height of salient cues, which probably provided increased contrast against the sky, was primarily relevant for homing [7, 10, 14, 19, 21, 22], while van Iersel and van den Assem [10] working with *Bembix rostrata*, using large flat rings, found surface area to be more important than height. Baerends [14] also believed that surface area of visual cues was an important property of landmarks used for homing. The large size of the corks and cubes used in the exploratory experiments reported here were probably preferred by the wasps to the smaller, less-stimulating naturally-occurring homing cues. Tinbergen [20, 41], working with *Philanthus triangulum*, found that wasps preferred to follow larger cues within the burrow vicinity. The local cues introduced in the present experiments also had sharp corners (cubes) and edges (corks), and often contrasted with the light sand. Apart from angular size, height, texture and contrast, there is yet another aspect that makes objects close to the goal visually salient: as a homing insect moves closer to the goal, the images of close objects move faster across its visual field than those of more distant objects [e.g. 26, 61,62,63]. One problem with relying on local visual cues is that they are easily disturbed by natural events. For instance, one of us has witnessed at least seven *M. monodonta* to search for hours within their burrow vicinities when pock marks left on the sand surface after a night of heavy rain had altered local cues [22, p 23]. Their numerous digging attempts were clustered within limited areas, likely the result of the wasps' reliance on imprecise distant cues which brought them into the burrow vicinity. Baerends [14] commented that *Ammophila campestris* also reacted to rain-pocked surfaces and Tsuneki [19] observed that *Bembix niponica* searched for her burrow hours after it had been trampled. These observations indicate that when the original local cues have been disturbed, the wasps would use the remaining visual panorama to attempt to locate their burrow. The wider panorama, however, does not provide the precise homing data local cues do, resulting in many attempted diggings within the broad area prescribed by the distant cues. Persistence, along with trial and error, enable the wasps to eventually locate their burrows. Van Iersel and van den Assem [10] attributed this to a shift of attention from disturbed parts of the scene to the undisturbed part.

## The relationship between local and distant cues

Visual barriers inhibited most *M. monodonta* from locating their burrows, showing that distant visual cues were necessary for accurate homing. Van Iersel and van den Assem [10] demonstrated that this is also true for *Bembix rostrata*. They conducted a barrier experiment in which they placed a visual barrier, 120 cm x 50 cm, 60 cm behind the burrow of a homing wasp. The barrier did not prevent the wasp from locating her burrow, but did evoke persistent searching. When a 30 cm × 50 cm window, through which the wasp was able to see part of the frontal horizon (usually tree tops contrasting with the sky 80 m distant), was opened, the result was decreased searching time; in many cases the wasp located her burrow immediately. This demonstrated that distant cues offering additional information will enable a wasp to home more efficiently.

In the experiment where a *M. monodonta* wasp chose to dig at the ends of the barrier (Fig 4B and 4C), distant visual cues brought her to the burrow vicinity. Once she was there, the barrier masked much of the frontal panorama she had learned. However, distant cues at the left and right ends of the barrier were available and used by the wasp, though not successfully, since the wasp had already passed the latitude of her burrow entrance. Thorpe [16] conducted an experiment with *Ammophila pubescens* and observed similar results. He placed a barrier in front of a wasp dragging a caterpillar, the ends of the barrier equidistant from her path. Approaching the barrier, she deviated from her path toward the barrier's right end, then resumed her straight path. When placed before her again; she deviated toward the right end again, and the next time she chose the left end in her attempts to match the image she had learned. Both *M. monodonta* and *A. pubescens* used distant cues that were available to them, though not perfect matches to the images they learned to follow.

Experiments 1 and 2 show that *M. monodonta* are guided by cues within their burrow vicinities (local cues) to the location of their invisible burrow entrances. However, the barrier/cube-configuration experiment demonstrated that these local cues are effective only if distant cues are available. When the barrier was presented, although all 10 wasps were able to home to the vicinities of their burrows (guided by frontal, and perhaps lateral, distant cues), they were unable to locate their burrows. They were unable to see the distant cues because of the masking barrier; this caused a major disturbance leading to disorientation. This major disturbance might explain why they did not attempt to dig at the ends of the barrier, as the wasp in Fig 4B and 4C had, and why they appeared to be lost and "frantically" searching; they might not even have recognized that they were in their burrow vicinities. This experiment suggests that local and distant cues are used simultaneously [13] and not successively (as suggested by Plowright et al. [64]) as the wasps were unable to home in the absence of distant cues despite the presence of prominent local cues.

## Excluding non-visual local cues

The last set of experiments were designed to exclude non-visual cues in the burrow vicinities, targeting potential olfactory, but also tactile and auditory cues. Chmurzynski [21] believed that *Bembix rostrata* locating her burrow within 3 cm of the entrance were able to tactically sense the looser sand near the burrow [21, p 120]. Tinbergen also suggested *P. triangulum* used their sense of touch to detect the loose sand at the burrow entrance. Regarding auditory cues, the possibility of wasps responding to them might apply to species in which the sounds of larval movements might be detected, but this would probably not occur in *M. monodonta* since the egg does not hatch during provisioning. However sonar is a possibility if *M. monodonta* were to respond to changes in air-flow or wing-beat sound reflections due to surface density changes between the loose sand over the hollow burrow and the compact sand surrounding it. In any case, the inverted cover precluded this.

The wasps were able to home very close to their hidden burrows, using visual cues alone, demonstrating the surprising accuracy with which the wider panorama allowed the wasps to pinpoint their burrow location. The wasps were able to dig within a distance of 6 cm of the hidden burrow entrance, usually much less, with median distances in both x- and y-directions being less than 1 cm (Fig 6H). A persistent wasp could dig within the tight cluster of points determined by distant cues, despite the lack of local homing cues (Fig 2R). Tinbergen [20] had removed all local cues around the burrow of *Philanthus triangulum* while she was away foraging. Upon her return she was highly disturbed, but soon calmed down and made apparent trial and error diggings at the approximate site of her burrow entrance (relying on remaining visual cues), eventually succeeding. Under normal circumstances with moderate local disturbances, *M. monodonta*'s "trial and error" diggings are likely to eventually lead them to their burrow entrances also.

The search distributions across the different conditions in which the burrow was covered with different layers of material did not differ significantly. Wasps in all conditions tended to dig at locations in front of their burrow entrances (Fig 6H, top left panel), possibly a sign of uncertainty and a precaution against overshooting the goal. We cannot offer a lucid interpretation for the overall difference in search spread in x- and y-direction.

A final aspect worth considering is the question why ground-nesting wasps such as *M. monodonta* do not mark or appear to mark their burrow entrances using pheromones. One answer is likely to be the presence of parasites, which may also be the reason for some of the wasps carefully closing and camouflaging their burrow entrances [54]. Compared to olfaction, vision offers private information to burrow location that cannot be penetrated by a parasite. Another reason may also be that the loose, wind-shifting sand surface of the wasps' preferred nesting sites makes long-term pheromone marking difficult.

## Conclusion

Homing behavior in *M. monodonta* includes searching for and finding the burrow vicinity within a broad area by using a panorama of distant and local visual cues. The latter enable the wasps to precisely locate their invisible burrow entrances. There may thus be no functional distinction between homing to the broad burrow vicinity (although this remains to be investigated) and recognition of the burrow location as such: The wasps used the entire panorama, with distant and local cues attended to simultaneously. When local cues have changed, the wasps conduct apparently-random digging attempts within the 'catchment area' of remaining cues, and eventually locate their burrow entrances by trial and error. Supporting the simultaneous use of local and distant cues is the observation that when distant cues are disturbed (as when masked by a visual barrier) the wasps are unable to home, despite the presence of prominent local cues previously used. Furthermore, no cues other than visual are necessary for the precise pinpointing of the burrow location. The results reported here support the hypothesis of van Iersel and van den Assem [10] that homing digger wasps use distant and local visual cues simultaneously in order to find their burrows and are in full agreement with recent experimental and modelling evidence on the visual information used by insects when pinpointing goals (reviewed in [65]).

## Supporting information

**S1 Fig. Top: Example of the visual scene at the field site. Bottom: Scene low-pass filtered to approximately 1deg resolution.**
(TIF)

**S2 Fig. Top: Example of the visual features at the site of a barrier experiment. Bottom: The scene a wasp is confronted with when the barrier is in place. Arrow points to the wasp's burrow.**
(TIF)

**S1 Data. Excel spreadsheet containing data used for Fig 6H: x/y position of means of search distributions relative to the true nest location for 12 wasps for each of the conditions A-G.** Same for the spread of search distributions.
(XLSX)

## Acknowledgments

MC wishes to thank the late Dr. J. J. van Iersel for his encouragement, enlightening discussions, and sharing his unpublished results; his dear wife Grace Donaldson Cormons for reading and discussing many drafts; and cartographer Bill Nelson for updating original figures from MC's masters thesis. We are grateful to Tom Collett and Antoine Wystrach for their comments and suggestions on an earlier draft of the manuscript and both Femke Batsleer and Volker Nehring for their meticulous reviewing. We are in particular grateful to Jan Hemmi for statistical analysis advice, design and code.

## Author Contributions

**Conceptualization:** Matthew J. Cormons.

**Data curation:** Matthew J. Cormons.

**Formal analysis:** Matthew J. Cormons.

**Investigation:** Matthew J. Cormons.

**Methodology:** Matthew J. Cormons.

**Project administration:** Matthew J. Cormons.

**Resources:** Matthew J. Cormons.

**Validation:** Matthew J. Cormons.

**Visualization:** Matthew J. Cormons, Jochen Zeil.

**Writing – original draft:** Matthew J. Cormons.

**Writing – review & editing:** Matthew J. Cormons, Jochen Zeil.

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
