## [Decision Letter · Decision Letter 0]

23 Feb 2022

PONE-D-21-38862THE HOMING BEHAVIOR OF THE DIGGER WASP MICROBEMBEX MONODONTA SAY (HYMENOPTERA, CRABRONIDAE)PLOS ONE

Dear Dr. Zeil,

Thank you for submitting your manuscript to PLOS ONE. After careful consideration, we feel that it has merit but does not fully meet PLOS ONE’s publication criteria as it currently stands. Therefore, we invite you to submit a revised version of the manuscript that addresses the points raised during the review process.

The two reviewers pointed out a large number of things that have to be considered. This means that the paper will have to be rethought and rewritten according to the referees comments and suggestions. 

We look forward to receiving your revised manuscript.

Kind regards,

Nicolas Chaline

Academic Editor

PLOS ONE

Journal Requirements:

3. Please amend your authorship list in your manuscript file to include author Jochen Zeil

Reviewers' comments:

Reviewer's Responses to Questions

**Comments to the Author**

1. Is the manuscript technically sound, and do the data support the conclusions?

Reviewer #1: Partly

Reviewer #2: Partly

2. Has the statistical analysis been performed appropriately and rigorously? 

Reviewer #1: N/A

Reviewer #2: No

3. Have the authors made all data underlying the findings in their manuscript fully available?

Reviewer #1: No

Reviewer #2: No

4. Is the manuscript presented in an intelligible fashion and written in standard English?

Reviewer #1: Yes

Reviewer #2: Yes

5. Review Comments to the Author

Reviewer #1: The authors present a series of inventive experiments to research the cues (local, distant, possible non-visual cues) related to nest recognition in the digger wasp Microbembix monodonta. I appreciate that the authors wrote this manuscript from data collected almost 50 years ago. I admire the perseverance and courage needed to do this and it deserves a place in the scientific literature.

However, the manuscript needs profound rewriting/restructuring, rethinking of term use and reframing of the experiments in light of the current scientific literature, as the research is somewhat outdated. Nevertheless, if the manuscript is rewritten, this would be a nice natural history publication. I see room for turning the disadvantage of reporting old experiments into an advantage by highlighting the need for natural history research to set the base for current behavioral and ecological research. This type of research is becoming rare but remains valuable. I really want to motivate the authors to rewrite their manuscript because renewed interest for this kind of work is needed, but in an accessible format for current researchers.

The data is quite hidden in text and figures with no clear overview of the data itself. The data should be made available in a more direct format (e.g. tables).

In depth suggestions and comments are given below.

I am looking forward to give more detailed comments on a revised manuscript.

Major comments:

I find the term use ‘homing behavior’ too broad for how it is used in the manuscript. This probably stems from how this term was used in for instance Tinbergen’s papers, but its meaning has expanded in the meantime. Homing behavior is currently used mainly for large distance orientation back to a nest site (Schöne & Tengö, 1991; Tengö et al., 1996; Goulson & Stout, 2001; also see ornithological research). Nest recognition (see for instance Plowright et al., 1995) can be seen as part of this, but they are not synonyms. I advise to rewrite the manuscript by being more specific in what is studied in this manuscript, for instance using the term ‘nest recognition’. Homing behavior should then be added to the key words.

Title: related to the previous comment, I would use something more specific and a title that says something about the results of the manuscript rather than a title that is a very broad topic in itself. A suggestion: ‘The digger wasp Microbembix monodonta uses visual cues in nest recognition’.

Abstract: I think there are too many details here regarding the methods and experiments. As a reader without any background on the story, I was a bit lost regarding the bigger picture and where this was headed. Shorten the abstract and mention what type of experiments were performed, but don’t give details how exactly. Focus on the outcome and interpretation of the results. Currently, methods and results and interpretation are presented mixed. Make a flow more parallel to the paper (intro-methods-results-interpretation/conclusion) in the abstract as well. Some suggestions: L14-L19, shorten this into one or two sentences with, for instance, following structure: “Returning wasps were monitored where they attempted to dig when either one or a combination of local cues and visual barriers were introduced.” L19-26: shorten this likewise with the potential olfactory, auditory and tactile homing cues as focus. Then in a next sentence, summarize the results and then give your conclusion.

The writing is in a style that is very focused on the author (sentences often start with ‘I…’ and ‘my…’). In general, I do like a more active style in manuscripts, as writing too passively often makes it more boring. However, here, it is often distracting from the main topic or experiments. I advise to rewrite such sections by putting the stress on the subject of the experiments rather than the author who performed something. (See for instance: L45, L55, L60, most subsections in the methods,… L257,…).

There is a lack in discussing this research within a recent scientific framework. How is this research relevant for current behavioral and movement ecology, evolutionary biology,…? Both introduction and discussion could use embedding in a more modern framework. I think a good starting point to stress the value of your work would be in the importance of natural history in current ecological and evolutionary research, see for instance Travis (2020).

It should be more clear in the introduction (or methods) what the exact aim is of the visual barriers: is it to obstruct local cues so they should only rely on more distant cues? Or rather that they only see the local ones when they are already close to the nest? This should be explained somewhere in more detail.

The results are quite hidden in the text and figures. It might be useful to give results in a table (how many wasps of the total had what response; even if it is one row for one table). Especially because now I have to look back and forth for the number of wasps for a certain response and total numbers. And such results are in need of some basic statistics: chi-squared tests should give a basic idea if the number of wasps is different for the different response groups. These statistics could be put in the same table.

Discussion: a lot of new context or terms are introduced here (e.g. retinotopic matching, the use of local and distances cues). I would suggest to already move some of these things to the introduction, to make the introduction and discussion work in parallel. As the introduction is now very short, some shifting of the body from the discussion to the introduction would balance this more.

Structure of discussion: in general, the discussion feels quite chaotic to me and mainly has to do with the writing structure and the style (e.g. starting a new paragraph with ‘also’, ‘with regard to’). I suggest to try to integrate paragraphs more, start the paragraph with a sentence that already gives the focus of that paragraph rather than the context (which should be described in the body). Such a structure is more readable for a first reader, especially also for someone who is scanning the paper for interesting discussion topics (they will mainly look at the first and last sentence of each paragraph to find what is useful for them). For instance, paragraph 2 of the discussion: start with a sentence regarding retinotopic matching; paragraph 3: disorientation; suggestion: then a third paragraph on local and distant cues,…

Conclusion: new discussion topics are brought up (around L382, and especially from L385 onward), which should be integrated in a paragraph in the main discussion regarding distant and local cues, rather than starting a new discussion topic in the conclusion.

Writing style in general: it feels a bit outdated, but you can use for instance Plowright et al (1995) as an example regarding general structuring. I don’t see ‘being outdated’ as something negative, it is what it is (I actually also like it in a way when reading manuscripts from the ’60, ’70 etc). But I am convinced the manuscript would be more accessible to, and picked up more by, current researchers if a more ‘modern’ writing style would be adopted. With that, I mainly mean how an abstract is structured, writing style not focused on the first person form, how the hypotheses or research questions are introduced in the introduction, what the first paragraph of an introduction does, etc…

Minor comments:

L14: accomplishes

L45: avoid literally referring to ‘my master’s thesis’, as the manuscript here should be an independent report of the results. I suggest to rewrite this sentence to focus on what is actually done and the research question (with a reference to the thesis).

L49-50: ‘rendering it invisible’: this feels quite anthropomorphic (in the sense that is invisible to us), but probably with the main function to be invisible to possible predators/parasites (see Poldori et al. (2009) and references therein). As you describe experiments on how it is not (indirectly) invisible to the wasps themselves, I would reformulate this sentence by focusing on the function of closing their nests.

L61-63: I would omit this here, and rather give a clear summary of research questions + short what was done in the last paragraph of the introduction.

L63: omit or reformulate (suggestion: ‘Three types of experiments were performed to test the influence of local visual cues, the influence of distant visual cues, the combination of the former two and the influence of non-visual cues.’ Then the rest of the methods follows, but it is already clear for the reviewer what structure or subsections to expect.)

L120 and following + fig 1: can you give more meaningful names to the conditions? There are quite a few conditions, and when I try to understand table 1 I constantly have to go back and forth between table and main text. Maybe use the ‘acrylic-paper-cloth-sand’ wording? It’s longer, but more meaningful and would make table 1 more directly interpretable.

Table 1: this table can use more explanation in the caption (a summary of what is in the text). A table (and figure) should be interpretable on its own without having to read the full text (+ they are the billboards for your paper).

Fig.4: C should be given in a table (see previous comment on results in general), then it would be more clear what the total number of wasps were.

Fig.5: mention somewhere in the caption what the actual nest is (the yellow dot I presume?).

L257-263. Give a summary of all the results and only start the discussion in the next paragraph, as L260-263 is already a very specific discussion point in itself, and should be integrated elsewhere. The first paragraph of the discussion should be a summary of the results.

L337: ‘and those of others with digger wasps’: give references.

L358: ‘Conditions (F) and (G)’: give more meaningful explanation, as someone reading the discussion might not have read the methods in detail.

L382: no closing bracket present.

L391: I like this concluding interpretation/sentence!

References

David Goulson and Jane C. Stout (2001) Homing ability of the bumblebee Bombus terrestris (Hymenoptera: Apidae) Apidologie, 32 1 (2001) 105-111. DOI: https://doi.org/10.1051/apido:2001115

Polidori, C., Ouadragou, M., Gadallah, N.S. & Andrietti, F. (2009) Potential role of evasive flights and nest closures in an African sand wasp, Bembix sp. near capensis Lepeletier 1845 (Hymenoptera Crabronidae), against a parasitic satellite fly. Tropical Zoology 22: 1-14

Schöne, H. & Tengö, J. (1991). Homing in the Digger Wasp Bembix rostrata - Release Direction and Weather Conditions. Ethology, 87, 160–164.

Tengö, J., Schöne, H., Kühme, W., Schöne, H. & Kühme, L. (1996). Nesting cycle and homing in the digger wasp Bembix rostrata. Ethol. Ecol. Evol., 8, 207–211.

Joseph Travis (2020) Where Is Natural History in Ecological, Evolutionary, and Behavioral Science? The American Naturalist 196:1, 1-8, DOI: 10.1086/708765

C M S Plowright, Colleen E O'Connell, Lisa J Roberts & Sheri L Reid (1995) The use of proximal and distal cues in nest entrance recognition by bumble bees, Journal of Apicultural Research, 34:2, 57-64, DOI: 10.1080/00218839.1995.11100888

Reviewer #2: The present manuscript by Mathew Cormons describes a series of experiments designed to determine the cues used by the digger wasp Microbembex monodonta to determine the position of its nest. Although the study seems interesting, and the experiments are sound and have been well designed, I am not positive about the news-value of the study or whether it just adds another digger wasp species to the wide literature on homing behaviour in digger wasps. This is mainly due to the fact that the author does not justify his experiments (which could be done in the introduction) to convince the reader that they are really necessary. Another point of criticism concerns the analytical part of the study. Although I guess that many things can be explained without an overwhelming amount of tricky statistical tests, the results need to be quantified in some way and not remain a mere qualitative description of what happened. I am sure the author will be able to cope with these concerns in his revision of the text.

Comments to the authors

(1) You need to use your introduction to justify your experiments. Is there any evidence in the literature that the wasps might use other cues than visual surface structures for their homing behaviour. I could accept chemical cues (maybe brood scent), but auditory (from what?) and tactile cues (differences in sand structure?) seem a bit far fetched. But since you planned and executed the experiments, you might have some reasonable explanation. Please explain better why you think these cues are worth testing in your introduction.

(2) You say that the opaque barrier blocked distant visual cues. What were these distant cues? Please provide evidence that the wasps actually see these "distant cues" (depending on eye-morphology, some insects are very short-sighted). Also interesting in this context is that they nest in "sandy areas with sparse vegetation" (line 47) and that your "study area was a mostly-bare, sandy blowout" (line 55). Thus distant cues might be extremely distant. Couldn't it be that the barrier just provided an additional landscape cue that the wasps did not "understand"? Alternative to your interpretation, the wasps might use solar compass (rough navigation to approach the site) and landscape structures for fine tuning of their homing and, eventually, surface cues for identifying the exact position. The new landscape cue you introduced might just have caused a general confusion in the wasps, and they abandoned the search. However, they should be used to changes in surface cues, since sand ripples are bound to change over time. Maybe you could discuss this topic.

(3) You need to quantify your results. In your experiments on the exclusion of non-visual cues (lines 101-154) you show how this can be done. Why didn't you do the same (or something similar) in the other experiments? With these data you could even perform statistics (correct for pseudoreplication since you evaluated multiple events of the same wasp), which would be interesting particularly when looking at the digging attempt distribution of "olfactory cues only" against "visual surface cues only". There could actually be a statistical difference between these conditions.

(4) What was the purpose of Experiment 2 (lines 161 ff), which just repeats what you already tested before in Experiment 1 (I got a bit confused with the labelling of the conditions in the text, but in Figure 6 it is Fig. 6C and Fig.6F). Did you expect something "new" or different from this experiment? If so, why? If not, why did you perform the experiment?

6. PLOS authors have the option to publish the peer review history of their article (what does this mean?). If published, this will include your full peer review and any attached files.

Reviewer #1: **Yes: **Femke Batsleer

Reviewer #2: No

---

## [Author Response · Author response to Decision Letter 0]

12 Apr 2022

We thank reviewers for their constructive and positive comments. In this revision, we took most of their suggestions on board, which we detail below. In particular:

(1) We changed the title (ref. 1).

(2) We shortened the abstract to concentrate on outcomes and interpretation of results (ref. 1).

(3) We completely reworked the introduction and the discussion to put our results into the perspective of recent work on insect visual navigation, homing and nest entrance localization (ref. 1).

(4) We added statistics to the outcomes of the non-visual cue experiments (ref. 2) 

Detailed responses to reviewer comments:

Reviewer #1: The authors present a series of inventive experiments to research the cues (local, distant, possible non-visual cues) related to nest recognition in the digger wasp Microbembix monodonta. I appreciate that the authors wrote this manuscript from data collected almost 50 years ago. I admire the perseverance and courage needed to do this and it deserves a place in the scientific literature.

However, the manuscript needs profound rewriting/restructuring, rethinking of term use and reframing of the experiments in light of the current scientific literature, as the research is somewhat outdated. Nevertheless, if the manuscript is rewritten, this would be a nice natural history publication. I see room for turning the disadvantage of reporting old experiments into an advantage by highlighting the need for natural history research to set the base for current behavioral and ecological research. This type of research is becoming rare but remains valuable. I really want to motivate the authors to rewrite their manuscript because renewed interest for this kind of work is needed, but in an accessible format for current researchers.

Response: Thank you and we appreciate your thoughts here. Rather than highlighting ‘natural history’ (which we do think is very important and much neglected) we rather chose to frame the experiments in the context of recent research in insect visual navigation, so more in a neuroethological context.

The data is quite hidden in text and figures with no clear overview of the data itself. The data should be made available in a more direct format (e.g. tables).

Response: We now make the data for the non-visual cue experiments available as an Excel spread sheet.

In depth suggestions and comments are given below.

I am looking forward to give more detailed comments on a revised manuscript.

Major comments:

I find the term use ‘homing behavior’ too broad for how it is used in the manuscript. This probably stems from how this term was used in for instance Tinbergen’s papers, but its meaning has expanded in the meantime. Homing behavior is currently used mainly for large distance orientation back to a nest site (Schöne & Tengö, 1991; Tengö et al., 1996; Goulson & Stout, 2001; also see ornithological research). Nest recognition (see for instance Plowright et al., 1995) can be seen as part of this, but they are not synonyms. 

I advise to rewrite the manuscript by being more specific in what is studied in this manuscript, for instance using the term ‘nest recognition’. Homing behavior should then be added to the key words.

Response: We use the term ‘pinpointing the nest entrance’ now to be clearer and in line with the terminology used in the more recent insect navigation literature. We have reworked both abstract and the introduction to put our work in the context of recent research in this area (details below).

Title: related to the previous comment, I would use something more specific and a title that says something about the results of the manuscript rather than a title that is a very broad topic in itself. A suggestion: ‘The digger wasp Microbembix monodonta uses visual cues in nest recognition’.

Response: Title now reads: DIGGER WASPS MICROBEMBEX MONODONTA SAY (HYMENOPTERA, CRABRONIDAE) RELY EXCLUSIVELY ON VISUAL CUES WHEN PINPOINTING THEIR NEST ENTRANCES.

Abstract: I think there are too many details here regarding the methods and experiments. As a reader without any background on the story, I was a bit lost regarding the bigger picture and where this was headed. 

Shorten the abstract and mention what type of experiments were performed, but don’t give details how exactly. Focus on the outcome and interpretation of the results. Currently, methods and results and interpretation are presented mixed. Make a flow more parallel to the paper (intro-methods-results-interpretation/conclusion) in the abstract as well. 

Some suggestions: L14-L19, shorten this into one or two sentences with, for instance, following structure: “Returning wasps were monitored where they attempted to dig when either one or a combination of local cues and visual barriers were introduced.” L19-26: shorten this likewise with the potential olfactory, auditory and tactile homing cues as focus. Then in a next sentence, summarize the results and then give your conclusion.

Response: Done.

The writing is in a style that is very focused on the author (sentences often start with ‘I…’ and ‘my…’). In general, I do like a more active style in manuscripts, as writing too passively often makes it more boring. However, here, it is often distracting from the main topic or experiments. 

I advise to rewrite such sections by putting the stress on the subject of the experiments rather than the author who performed something. (See for instance: L45, L55, L60, most subsections in the methods,… L257,…).

Response: We followed the advice throughout with a few exceptions, were an active form was more appropriate. We are now also co-authoring the paper.

There is a lack in discussing this research within a recent scientific framework. How is this research relevant for current behavioral and movement ecology, evolutionary biology,…? Both introduction and discussion could use embedding in a more modern framework. I think a good starting point to stress the value of your work would be in the importance of natural history in current ecological and evolutionary research, see for instance Travis (2020).

Response: We now review current concepts and research background on insect visual navigation, including pinpointing goals, such as nests, in the introduction and discuss our results within the current frame work in addition to the extensive older literature.

It should be more clear in the introduction (or methods) what the exact aim is of the visual barriers: is it to obstruct local cues so they should only rely on more distant cues? Or rather that they only see the local ones when they are already close to the nest? This should be explained somewhere in more detail.

Response: We now clarify in the introduction (line 48ff): “ Cartwright and Collett [25,42] have shown that this search behaviour can be explained by assuming that insects memorize the view from the nest entrance (as if taking a snapshot) and upon returning to the nest area moving in such a way as to minimize the mismatch between what they currently see and their memorized snapshot. This snapshot model of homing can be generalized to global image matching without identification of individual objects in a panoramic view from the nest (3,5,43). The crucial insight here was that a location in a natural environment is uniquely defined by the view taken from it because global image differences – in the simplest way measured as the root mean squared pixel difference between a memorized goal image and the images seen on approach to the goal – become systematically smaller as the distance to the goal decreases. Pinpointing the nest entrance, thus becomes a gradient descent in global image differences [43,44, see also 45-51] and experiments with large screens have added to other evidence (see examples in [1]) that navigating insects do make use of the full visual panorama and not only of individual landmarks [9].”

The results are quite hidden in the text and figures. It might be useful to give results in a table (how many wasps of the total had what response; even if it is one row for one table). Especially because now I have to look back and forth for the number of wasps for a certain response and total numbers. And such results are in need of some basic statistics: chi-squared tests should give a basic idea if the number of wasps is different for the different response groups. These statistics could be put in the same table.

Response: (1) We now clarify in the text and in figure legends that Fig. 3 shows individual examples of outcomes observed in x number of wasps (2) We added numbers to the bar graph in Fig. 4C and (3) now show the results of the non-visual cue experiment in the form of notched boxplots in Fig. 5H together with wasp and sample numbers, which offers an immediate visual assessment of significant differences in the search distributions. This was also requested by refer 2. We also re-drew the side-view insets to improve clarity.

Discussion: a lot of new context or terms are introduced here (e.g. retinotopic matching, the use of local and distances cues). I would suggest to already move some of these things to the introduction, to make the introduction and discussion work in parallel. As the introduction is now very short, some shifting of the body from the discussion to the introduction would balance this more.

Response: We introduce current concepts of visual navigation based on image matching in the introduction including open questions regarding the role of local and global cues in pinpointing goals. We then can refer back to these concepts in the discussion of our results.

Structure of discussion: in general, the discussion feels quite chaotic to me and mainly has to do with the writing structure and the style (e.g. starting a new paragraph with ‘also’, ‘with regard to’). 

I suggest to try to integrate paragraphs more, start the paragraph with a sentence that already gives the focus of that paragraph rather than the context (which should be described in the body). Such a structure is more readable for a first reader, especially also for someone who is scanning the paper for interesting discussion topics (they will mainly look at the first and last sentence of each paragraph to find what is useful for them). For instance, paragraph 2 of the discussion: start with a sentence regarding retinotopic matching; paragraph 3: disorientation; suggestion: then a third paragraph on local and distant cues,…

Response: We now begin the discussion with a summary of results and a brief introduction into the issues to be discussed. We have substantially reworked the discussion to create a more logical flow.

Conclusion: new discussion topics are brought up (around L382, and especially from L385 onward), which should be integrated in a paragraph in the main discussion regarding distant and local cues, rather than starting a new discussion topic in the conclusion.

Response: Done.

Writing style in general: it feels a bit outdated, but you can use for instance Plowright et al (1995) as an example regarding general structuring. I don’t see ‘being outdated’ as something negative, it is what it is (I actually also like it in a way when reading manuscripts from the ’60, ’70 etc). But I am convinced the manuscript would be more accessible to, and picked up more by, current researchers if a more ‘modern’ writing style would be adopted. With that, I mainly mean how an abstract is structured, writing style not focused on the first person form, how the hypotheses or research questions are introduced in the introduction, what the first paragraph of an introduction does, etc…

Response: We hope that the revised manuscript does now take these issues into account.

Minor comments:

L14: accomplishes

Response: Removed.

L45: avoid literally referring to ‘my master’s thesis’, as the manuscript here should be an independent report of the results. I suggest to rewrite this sentence to focus on what is actually done and the research question (with a reference to the thesis).

Response: Removed.

L49-50: ‘rendering it invisible’: this feels quite anthropomorphic (in the sense that is invisible to us), but probably with the main function to be invisible to possible predators/parasites (see Poldori et al. (2009) and references therein). As you describe experiments on how it is not (indirectly) invisible to the wasps themselves, I would reformulate this sentence by focusing on the function of closing their nests.

Response: Wording changed (line 74): “…then kick sand over the burrow entrance, making it visually impossible to locate, certainly to a human investigator and probably to the wasp herself, as well as to potential predators and parasites.”

L61-63: I would omit this here, and rather give a clear summary of research questions + short what was done in the last paragraph of the introduction.

L63: omit or reformulate (suggestion: ‘Three types of experiments were performed to test the influence of local visual cues, the influence of distant visual cues, the combination of the former two and the influence of non-visual cues.’ Then the rest of the methods follows, but it is already clear for the reviewer what structure or subsections to expect.)

Response: The second paragraph of the Method section now reads (Line 93ff): “The following experiments were conducted to test for: 1) the influence of local visual cues, 2) the influence of distant cues, 3) the interaction between local and distant cues, and 4) the influence of non-visual cues.”

L120 and following + fig 1: can you give more meaningful names to the conditions? There are quite a few conditions, and when I try to understand table 1 I constantly have to go back and forth between table and main text. Maybe use the ‘acrylic-paper-cloth-sand’ wording? It’s longer, but more meaningful and would make table 1 more directly interpretable.

Response: We now follow the referee’s advice on labelling the conditions.

Table 1: this table can use more explanation in the caption (a summary of what is in the text). A table (and figure) should be interpretable on its own without having to read the full text (+ they are the billboards for your paper).

Response: We have changed the title, the caption and the condition names of Table 1 as suggested. 

Fig.4: C should be given in a table (see previous comment on results in general), then it would be more clear what the total number of wasps were.

Response: We now added numbers of wasps above bars and total number in heading.

Fig.5: mention somewhere in the caption what the actual nest is (the yellow dot I presume?).

Response: Added to figure legend (line 306ff): “Burrow entrance marked with a white circle at the intersection of x- and y-axes as labeled in G.”

L257-263. Give a summary of all the results and only start the discussion in the next paragraph, as L260-263 is already a very specific discussion point in itself, and should be integrated elsewhere. The first paragraph of the discussion should be a summary of the results.

Response: Beginning of the discussion now reads (line 322ff): “The experiments presented here demonstrate that M. monodonta wasps very quickly learn to use introduced local visual cues to locate their burrows. When these cues are shifted the wasps’ first digging attempts were always at the location they were last successful relative to the configuration. However, when the more distant visual panorama was blocked by barriers, wasps were unable to locate their burrows, even when the local cues they previously followed were still present. Finally, wasps were able to locate their hidden burrow position to within a few centimeters when all possible non-visual local cues were excluded by different layers of materials covering the nest entrance, including an airtight cover. M. monotonta, like many other nesting insects, thus rely exclusively on local visual cues in the context of the wider visual panorama to locate their hidden nest entrances. 

 These results confirm a large body of literature on the homing abilities of insects, but also raise a number of interesting questions regarding the accuracy of visual homing and the relationship between global visual cues and the fine visual details close to the goal. Lastly, we discuss the question why other, especially olfactory cues do not play a role in nest entrance recognition.”

L337: ‘and those of others with digger wasps’: give references.

Response: Removed.

L358: ‘Conditions (F) and (G)’: give more meaningful explanation, as someone reading the discussion might not have read the methods in detail.

Response: Section now reads (line 441ff): “The search distributions across the different conditions in which the burrow was covered with different layers of material did not differ significantly except in the following cases: first, wasps searched closer to the burrow location in the direction of approach (y-direction) in Conditions E and F, where covers consisted of flexible cloth and of a flexible cloth over a flexible plastic sheet, respectively. Interestingly, these were the only conditions that hugged the sand surface (0 cm above the surface), indicating that the wasps may have payed attention to subtle differences in elevations of the sand surface. M. monodonta approach their burrow vicinity flying very low, about 1 cm above the sand surface, facing forward (toward the cell end of the burrow). Slight hills and valleys in the sand surface might help the wasps in precisely locating their burrow entrances. With the elevated covers in the other conditions, wasps were forced to fly higher than normal and may have not been able to detect elevation differences very well.”

L382: no closing bracket present.

Response: Removed.

L391: I like this concluding interpretation/sentence!

Response: Thanks, but this had to be moved in the revision to line 380ff.

References

David Goulson and Jane C. Stout (2001) Homing ability of the bumblebee Bombus terrestris (Hymenoptera: Apidae) Apidologie, 32 1 (2001) 105-111. DOI: https://doi.org/10.1051/apido:2001115

Polidori, C., Ouadragou, M., Gadallah, N.S. & Andrietti, F. (2009) Potential role of evasive flights and nest closures in an African sand wasp, Bembix sp. near capensis Lepeletier 1845 (Hymenoptera Crabronidae), against a parasitic satellite fly. Tropical Zoology 22: 1-14

Schöne, H. & Tengö, J. (1991). Homing in the Digger Wasp Bembix rostrata - Release Direction and Weather Conditions. Ethology, 87, 160–164.

Tengö, J., Schöne, H., Kühme, W., Schöne, H. & Kühme, L. (1996). Nesting cycle and homing in the digger wasp Bembix rostrata. Ethol. Ecol. Evol., 8, 207–211.

Joseph Travis (2020) Where Is Natural History in Ecological, Evolutionary, and Behavioral Science? The American Naturalist 196:1, 1-8, DOI: 10.1086/708765

C M S Plowright, Colleen E O'Connell, Lisa J Roberts & Sheri L Reid (1995) The use of proximal and distal cues in nest entrance recognition by bumble bees, Journal of Apicultural Research, 34:2, 57-64, DOI: 10.1080/00218839.1995.11100888

Reviewer #2: The present manuscript by Mathew Cormons describes a series of experiments designed to determine the cues used by the digger wasp Microbembex monodonta to determine the position of its nest. Although the study seems interesting, and the experiments are sound and have been well designed, I am not positive about the news-value of the study or whether it just adds another digger wasp species to the wide literature on homing behaviour in digger wasps. This is mainly due to the fact that the author does not justify his experiments (which could be done in the introduction) to convince the reader that they are really necessary. 

Response: We now justify the experiments in the introduction with reference to the recent literature and to current concepts in visual navigation in insects, including aspects such as precise goal localization which are still not adequately known.

Another point of criticism concerns the analytical part of the study. Although I guess that many things can be explained without an overwhelming amount of tricky statistical tests, the results need to be quantified in some way and not remain a mere qualitative description of what happened. I am sure the author will be able to cope with these concerns in his revision of the text.

Response: We now have added panel H to Fig. 5, which shows the search distributions as notched boxplots which allow the reader to quickly assess significant differences between the distributions.

Comments to the authors

(1) You need to use your introduction to justify your experiments. Is there any evidence in the literature that the wasps might use other cues than visual surface structures for their homing behaviour. I could accept chemical cues (maybe brood scent), but auditory (from what?) and tactile cues (differences in sand structure?) seem a bit far fetched. But since you planned and executed the experiments, you might have some reasonable explanation. Please explain better why you think these cues are worth testing in your introduction.

Response: We now provide this explanation in the introduction (line 57ff): “Pinpointing the nest entrance, thus becomes a gradient descent in global image differences [43,44, see also 45-51] and experiments with large screens have added to other evidence (see examples in [1]) that navigating insects do make use of the full visual panorama and not only of individual landmarks [9]. However, whether it is the increasing salience of local visual cues that guide the final approach to the nest or whether pinpointing the nest location requires an ‘attentional’ switch to fine visual details, such as the nest entrance itself or visual features around it, is not known at present.

Moreover, in most cases it remains also unclear to what extent other cues help ground-nesting insects in their final approach to the nest. Desert ants, for instance, are guided by the CO2 plume emanating from the nest entrance [52] and other, very close-range cues such as local surface topography or even the sound of moving larvae emanating from the nest may help ground-nesting insects to pinpoint the exact location of the entrance. The role of such non-visual cues may be particularly important for insects such as the Microbembex monodonta we studied here, which return to a nest the entrance of which they had previously carefully closed, covered and camouflaged.” 

(2) You say that the opaque barrier blocked distant visual cues. What were these distant cues? Please provide evidence that the wasps actually see these "distant cues" (depending on eye-morphology, some insects are very short-sighted). 

Also interesting in this context is that they nest in "sandy areas with sparse vegetation" (line 47) and that your "study area was a mostly-bare, sandy blowout" (line 55). Thus distant cues might be extremely distant. Couldn't it be that the barrier just provided an additional landscape cue that the wasps did not "understand"? 

Response: We now provide two additional figures in the supplementary material that show the wider visual environment at the study site (Fig. S1) and the visual background that was affected by the barrier in the experiment documented in Fig. 4. We also note that visual systems ’record’ all radiant (sky) and reflected light (terrestrial objects) irrespective of their distance. Insects are known to see the sun (ants, honeybees) and even the milkyway (dung beetles) which are both infinitely far away. We try to give an impression of what a wasp in our experiments might see of the scene around their nests by low-pass filtering views to approximately 1deg resolution (Fig. S1 & S2).

Alternative to your interpretation, the wasps might use solar compass (rough navigation to approach the site) and landscape structures for fine tuning of their homing and, eventually, surface cues for identifying the exact position. The new landscape cue you introduced might just have caused a general confusion in the wasps, and they abandoned the search. However, they should be used to changes in surface cues, since sand ripples are bound to change over time. Maybe you could discuss this topic.

Response: We now discuss the role of local and global visual cues in insect homing in the introduction and the discussion with reference to current research, in addition to the older literature (see response above), but also line 353ff in the discussion: “The accuracy with which panoramic views provide information on location depends on the distance of visual features [43,58] and on the sensitivity of insects to image differences. Close to the burrow, the image differences experienced by homing insects will depend on the visual salience of nearby objects, in terms of their angular size and contrast with the background.” and more extensively in line 385ff.

(3) You need to quantify your results. In your experiments on the exclusion of non-visual cues (lines 101-154) you show how this can be done. Why didn't you do the same (or something similar) in the other experiments? With these data you could even perform statistics (correct for pseudoreplication since you evaluated multiple events of the same wasp), which would be interesting particularly when looking at the digging attempt distribution of "olfactory cues only" against "visual surface cues only". There could actually be a statistical difference between these conditions.

Response: We show a bar graph for the barrier experiments in Fig. 4C and now have added a notched boxplot analysis for the search distributions in Fig. 5H for both the distribution of means and of the search spread.

(4) What was the purpose of Experiment 2 (lines 161 ff), which just repeats what you already tested before in Experiment 1 (I got a bit confused with the labelling of the conditions in the text, but in Figure 6 it is Fig. 6C and Fig.6F). Did you expect something "new" or different from this experiment? If so, why? If not, why did you perform the experiment?

Response: Justified in line 203ff: “The objective of this experiment was similar the previous, but offered an air-tight seal that robustly excluded both olfactory and auditory cues.”

---

## [Decision Letter · Decision Letter 1]

22 Jul 2022

PONE-D-21-38862R1DIGGER WASPS MICROBEMBEX MONODONTA SAY (HYMENOPTERA, CRABRONIDAE) RELY EXCLUSIVELY ON VISUAL CUES WHEN PINPOINTING THEIR NEST ENTRANCESPLOS ONE

Dear Dr. Zeil,

Thank you for submitting your manuscript to PLOS ONE. After careful consideration, we feel that it has merit but does not fully meet PLOS ONE’s publication criteria as it currently stands. Therefore, we invite you to submit a revised version of the manuscript that addresses the points raised during the review process.

Your MS has much improved. However I would ask for you to take extra care in considering the statistical questions, linked with the small sample size the referee pointed out, and modifying the MS accordingly. Also consider the suggestions for improvements in the clarity and presentation of the MS.

We look forward to receiving your revised manuscript.

Kind regards,

Nicolas Chaline

Academic Editor

PLOS ONE

Journal Requirements:

Reviewers' comments:

Reviewer's Responses to Questions

**Comments to the Author**

1. If the authors have adequately addressed your comments raised in a previous round of review and you feel that this manuscript is now acceptable for publication, you may indicate that here to bypass the “Comments to the Author” section, enter your conflict of interest statement in the “Confidential to Editor” section, and submit your "Accept" recommendation.

Reviewer #1: (No Response)

2. Is the manuscript technically sound, and do the data support the conclusions?

Reviewer #1: Yes

3. Has the statistical analysis been performed appropriately and rigorously? 

Reviewer #1: No

4. Have the authors made all data underlying the findings in their manuscript fully available?

Reviewer #1: Yes

5. Is the manuscript presented in an intelligible fashion and written in standard English?

Reviewer #1: Yes

6. Review Comments to the Author

Reviewer #1: The authors provide a series of interesting experiments to study the scale and type of cues used for nest recognition in the digger wasp Microbembix monodonta. I applaud the authors for their extensive revisions, which made the manuscript a focused and clear piece of writing.

I find the experiments and results inspirational (for instance to engage students with in a field course), and especially the results regarding the absolute need for a panoramic cue intrigues me.

I have several minor comments to pinpoint things that are unclear (or not directly clear) to me which need some clarification, or suggestions that would ease the reading for me (or possible typos). The part that deserves the most clarifications, although improved already a lot since the first draft, is the influence of non-visual cues (see comments below), especially what the a priori expectations are of the different materials in the conditions.

My major concern is the representation of the data and statistics (see below), which the authors should consider carefully and find a solution for, fitting the overall aim of their manuscript (I give some suggestions below).

I think the authors will swiftly be able to incorporate my comments and suggestions, as they already made big improvements and incorporated the previous review comments appropriately.

Major comment:

Although the authors partially met the previous review comments regarding statistics and data transparency, my concerns have not completely disappeared.

First, the data presented, with very low sample sizes and repeated measurements on individuals, is in my opinion only suitable for a qualitative description. The authors also mainly write in such an approach, but to be completely transparent about it, the abstract and introduction should also mention the study is qualitative and/or descriptive.

The statistics provided for testing of non-visual cues does not seem appropriate and does not provide real significances (notched boxplots are compared visually, which can only point towards a possible significant difference, especially when comparing 8 groups). At least a one-way ANOVA should be applied, with post-hoc tests to check which groups are different. Preferably, adding a random effect for the individuals tested (as the measurements are not independent). However, there is probably not enough power for the latter. Apart from doing a proper ANOVA, another solution would be to omit any attempt of statistics (also omit the boxplots or only boxplots without grey area or notches), and be purely descriptive in the results and discussion (the discussion regarding the flexible cloth and sand surface could then be retained, but the part about spread in y-direction L453-455 can be omitted). In my opinion, this part is also of minor importance compared to the next experiment (air-tight cover), which is the most conclusive experiment regarding other possible cues.

Minor comments:

L23-25: Make this sentence singular, as the species name is in singular (it also broadens the results). E.g. ‘Here we show that a ground-nesting wasp Microbembex monodonta locates its hidden burrow entrances with the help of… only if the view of the wider panorama is not blocked.

L35 (and many others): delete ‘e.g.’ when citing literature. You always cite examples and can never be complete in a strict sense. Thus, also omit on L39, 44, 48, 340-342, 459

L37 (and many more): I would try to avoid the term ‘utilize’ and replace it by ‘use’. Utilize holds more the notion of making effective use of objects or a process leading to a product, often beyond its original intended use. The term ‘use’ is more neutral when describing an organism doing something (less anthropocentric).

L41: can you sum up a few of the groups here (probably similar to L340-342 then). For instance: ‘Vision in homing and pinpointing nests have been reported for other Hymenoptera and insect groups, such as …’

L44: not clear what ‘Marchand in…’ means, now it looks like referring to Marchand in all the manuscripts listed there, if it is a chapter in only the first, I would adjust the reference itself into a reference to the book chapter (instead of the complete book). https://guides.library.uq.edu.au/referencing/apa6/book-chapter

L47: reformulate ‘shifted introduced cues’: when first reading the sentence I didn’t understand what was meant, as the introducing of the cue happens before the shifting. Suggestion: ‘wasps followed introduced cues when these were shifted,…’

L58: ‘large screens’: make clear that this is for obstruction of the landscape (and not in the sense of digital screens).

L65: CO2, 2 subscript, not superscript.

L68: do you have a reference for this auditory cue of moving larvae as a cue?

L71: species is singular, so formulate sentence as singular (‘…is a solitary digger wasp that nests in…)

L74: ‘visually impossible to locate… probably to the wasp itself’: I would omit ‘and probably to the wasp herself’, this is quite a big jump/assumption and very anthropocentric (‘vision’ is anyway something completely different for an insect than what we can sense). The whole manuscript is about nest recognition and how it is indirectly not invisible to an individual wasp (through cues).

L75: repeat reference 63 here, to state that closing burrows is an anti-parasite behavior.

L79-L83: an introduction should end in research questions and/or hypotheses, not a conclusion of the results. Reformulate to research questions.

L90: it’s actually quite important that longitude is West (so -89.579446). I suggest to write it as one of the following: (46.172763N, 89.579446W), (46.172763, -89.579446)

L104: ‘trained’: what does this actually refer to? If it is was is described in the previous graph, indicate there what the training phase is. Suggestion (if I am correct in interpreting what the training is): L100 rewrite as ‘Each wasp was allowed to return undisturbed, enter, and leave. This is the training phase of the wasp to the objects.’

L114: Refer to fig 2 here (if I’m right).

L124: make sure to split the sentences with commas if a part is some form of a dependent clause. It’s of course partially a personal taste, but I like it when the writer guides me in where to ‘pause’ when reading (you have to assume as a writer that readers are lazy). The pauses are often obvious to a writer, but this is not always so for a reader. I tried to indicate it in the comments where I think extra comma’s or hyphens (or rather em dashes) would have eased my reading: take these as suggestions. For instance, here, I would a comma between ‘foraging’ and ‘a triangular’. Please also keep this in mind when re-reading your manuscript to add these where confusion with clauses might be possible.

L127: I wonder if the wasp was not confused already then by the experiment, because you removed already the local cues she probably used to replace them by your own?

L132: add something to explain this results in 6cm to the left of the burrow.

Influence of non-visual cues, experiment 1: this part already improved a lot in clarity compared to the previous version. I have some additional comments/suggestions to add to the clarity:

- Already refer in the beginning of this section to table 1

- State in the beginning what the aim is (explore different types of possible cues?)

- State what your expectations are a priori for each of the different materials. This is somewhat deducible from table 1, but deserves more explicit, elaborate explanation in the text (maybe somewhere around L140?). This would also make it more clear, step by step, to the reader how the experiment tests the different types of cues.

- L150 and further: as in the explanation for ‘condition (A)’, state which cues are excluded/included for each condition. It is summarized in table 1, but some written out explanation in the text and captions should complement this.

- L159 and further: as previous, state the hypothesis or what is excluded/included and elaborate. It is summarized in table 1, but in the text you can then give more explanation to for instance, the question marks in the table.

- L163: plexiglass should be in italics

- L168: what is exactly the purpose of this hole?

- L160, 161: 256 or 49 3 cm squares are hard to imagine, I would prefer 16 by 16 (or 16×16) squares with sides of 3 cm. Same for 21 cm² on L161; L164: give exact dimensions of fiberboard frame; give diameter of what is cut from the centre (hole or square?); L174: 256 3cm squares: give the dimensions as well

- L160: 11.0 is also in inches? And could you rescale those to cmxcm?

- Table1: 1) the uncertain states deserve more explanation, as in a previous comment, explain more in the text. 2) for E: cloth alone hugging surface: why is olfactory ruled out for this one? Deduced from the other conditions, I thought this was the one making the olfactory possible. And why would the cloth be masking the auditory cue?

L176: omit ‘to me’

L182: omit ‘for me’

L203: repeat the objective here (as it was not explicitly stated for experiment 1). But I would not say it is similar to it, but rather an alternative to see if only visual cues are enough to pinpoint the nest.

L206: give exact dimensions instead of squared centimeters.

General for results: state at the beginning of each part shortly what was tested again (readers are lazy), either by explaining it in the first sentence or adding a shot title (rather than “experiment 1”). For instance, on L229: ‘Figure 3B show the responses of one of the wasps to visual barrier that blocks distant visual cues.’

L233: ‘some’ is quite vague, can you quantify?

L253: explain a bit more in the material and methods (with LXX) that you moved the barriers. Here, in the results section, it comes out of nowhere.

L285: omit ‘of the’ (ellipses)

L346-348: I don’t really understand the azimuth/translation explanation. What does this mean for the orientation of the wasp, does it first set the angle correct and then approaches the goal? Can you elaborate on this?

L349-352: is this then the generalized model (as mentioned in the introduction; ‘global image matching’)? Does this one then expand on the snapshot model (as mentioned in the introduction), or is it a contrasting/competing model? Some extra explanation what the difference between both is could be helpful here. Then also end this paragraph with stating that your results support the latter model more. Rewriting the part in the introduction and discussion regarding snapshot and global image matching might be needed to synchronize both parts.

L361: comma between ‘rings’ and ‘found’

L365: em dashes, or commas, after Tinbergen and between ‘triangulum’ and ‘found’.

L369: what is ‘motion parallax’? When introducing a new term, elaborate on what it means and why it is relevant for your study.

L393: ‘to home more easily/rapidly’?

L414: a dot to much at the end of the sentence.

L452: ‘in front’ instead of forward (I was misguided by the ‘forward’ term as I first interpreted this as an actual overshoot).

L455: I doubt you can talk here about ‘significantly’ different from A, as the ‘statistical’ comparison is only with A (see major comments); if the spread of B is indeed higher than others, my first idea would be to think about the multitude of material layers added to this condition (or the frame, the spread seems high for the conditions with the frame, eyeballing the ellipses in Fig. 5).

L467, 469, 475: replace utilize by use

L469: put ‘:’ after ‘as such’, because the next sentence explains the ‘no functional distinction’. I would also nuance this statement more, because you did not test homing to the broad burrow vicinity, but only that in the burrow vicinity, a panorama is crucial for locating the nest entrance (which is a very interesting thing in itself). So I would not make a conclusion stating that homing to the nest aggregate is functionally similar to local nest recognition, as the former has not been tested. It probably uses landscape features as well on a larger scale, but this has not been tested. I suggest to omit this statement, and as a replacement stress the importance of the surrounding landscape.

7. PLOS authors have the option to publish the peer review history of their article (what does this mean?). If published, this will include your full peer review and any attached files.

Reviewer #1: **Yes: **Femke Batsleer

---

## [Author Response · Author response to Decision Letter 1]

10 Sep 2022

Reviewer #1: The authors provide a series of interesting experiments to study the scale and type of cues used for nest recognition in the digger wasp Microbembix monodonta. I applaud the authors for their extensive revisions, which made the manuscript a focused and clear piece of writing.

I find the experiments and results inspirational (for instance to engage students with in a field course), and especially the results regarding the absolute need for a panoramic cue intrigues me.

I have several minor comments to pinpoint things that are unclear (or not directly clear) to me which need some clarification, or suggestions that would ease the reading for me (or possible typos). The part that deserves the most clarifications, although improved already a lot since the first draft, is the influence of non-visual cues (see comments below), especially what the a priori expectations are of the different materials in the conditions.

My major concern is the representation of the data and statistics (see below), which the authors should consider carefully and find a solution for, fitting the overall aim of their manuscript (I give some suggestions below).

I think the authors will swiftly be able to incorporate my comments and suggestions, as they already made big improvements and incorporated the previous review comments appropriately.

Response: We thank Femke Batsleer for her meticulous referee work and her constructive comments. We have taken all her comments and suggestions on board as outlined in detail below (line number references are to the TrackChange version of the revision), but would very much like to keep the descriptive statistics in Fig. 5H for the reasons listed below.

Major comment:

Although the authors partially met the previous review comments regarding statistics and data transparency, my concerns have not completely disappeared.

First, the data presented, with very low sample sizes and repeated measurements on individuals, is in my opinion only suitable for a qualitative description. The authors also mainly write in such an approach, but to be completely transparent about it, the abstract and introduction should also mention the study is qualitative and/or descriptive.

The statistics provided for testing of non-visual cues does not seem appropriate and does not provide real significances (notched boxplots are compared visually, which can only point towards a possible significant difference, especially when comparing 8 groups). At least a one-way ANOVA should be applied, with post-hoc tests to check which groups are different. Preferably, adding a random effect for the individuals tested (as the measurements are not independent). However, there is probably not enough power for the latter. Apart from doing a proper ANOVA, another solution would be to omit any attempt of statistics (also omit the boxplots or only boxplots without grey area or notches), and be purely descriptive in the results and discussion (the discussion regarding the flexible cloth and sand surface could then be retained, but the part about spread in y-direction L453-455 can be omitted). In my opinion, this part is also of minor importance compared to the next experiment (air-tight cover), which is the most conclusive experiment regarding other possible cues.

Response: We would very much like to keep Fig. 5H for the following reasons: (1) The main and surprising result of this experiment is that the wasps can pinpoint the location of the hidden burrow to within a 12cm radius, regardless of any manipulation of local cues. (2) This fact is visually obvious from the size and location of search distributions as indicated by the ellipses. (3) The boxplots show these distributions in a quantitative way as distance of means to the true burrow location (centre location of ellipses) and as the extent of spread (size of ellipses). (4) Notched box plots are an accepted and honest piece of descriptive statistics that allows for a quick visual assessment of the shape of distributions and their differences. (5) The main insight is that these distributions are essentially very similar and a testimony to the astounding accuracy of the wasps. It is practically impossible and not very informative to find functional interpretations for the slight differences we can observe (e.g. medians of y-direction distributions differ by less than 1cm! Medians of spread distributions differ by less than 6cm!). (6) The boxplots also allow us to make data available (Supplementary S3) that can be used for further statistical analyses, if anybody feels that there is more insight to be gained. 

Minor comments:

L23-25: Make this sentence singular, as the species name is in singular (it also broadens the results). E.g. ‘Here we show that a ground-nesting wasp Microbembex monodonta locates its hidden burrow entrances with the help of… only if the view of the wider panorama is not blocked.

Done

L35 (and many others): delete ‘e.g.’ when citing literature. You always cite examples and can never be complete in a strict sense. Thus, also omit on L39, 44, 48, 340-342, 459

Done

L37 (and many more): I would try to avoid the term ‘utilize’ and replace it by ‘use’. Utilize holds more the notion of making effective use of objects or a process leading to a product, often beyond its original intended use. The term ‘use’ is more neutral when describing an organism doing something (less anthropocentric).

Done

L41: can you sum up a few of the groups here (probably similar to L340-342 then). For instance: ‘Vision in homing and pinpointing nests have been reported for other Hymenoptera and insect groups, such as …’

Response: We removed reference to other insect groups at this point.

L44: not clear what ‘Marchand in…’ means, now it looks like referring to Marchand in all the manuscripts listed there, if it is a chapter in only the first, I would adjust the reference itself into a reference to the book chapter (instead of the complete book). https://guides.library.uq.edu.au/referencing/apa6/book-chapter

Response: Added page reference.

L47: reformulate ‘shifted introduced cues’: when first reading the sentence I didn’t understand what was meant, as the introducing of the cue happens before the shifting. Suggestion: ‘wasps followed introduced cues when these were shifted,…’

Done

L58: ‘large screens’: make clear that this is for obstruction of the landscape (and not in the sense of digital screens).

Done

L65: CO2, 2 subscript, not superscript.

Done

L68: do you have a reference for this auditory cue of moving larvae as a cue?

Response: No, but larvae are likely to move and eat and this MUST create vibrations and airborne sound. To our knowledge, this has not been investigated in ground-nesting wasps.

L71: species is singular, so formulate sentence as singular (‘…is a solitary digger wasp that nests in…)

Done

L74: ‘visually impossible to locate… probably to the wasp itself’: I would omit ‘and probably to the wasp herself’, this is quite a big jump/assumption and very anthropocentric (‘vision’ is anyway something completely different for an insect than what we can sense). The whole manuscript is about nest recognition and how it is indirectly not invisible to an individual wasp (through cues).

Done.

L75: repeat reference 63 here, to state that closing burrows is an anti-parasite behavior.

Done.

L79-L83: an introduction should end in research questions and/or hypotheses, not a conclusion of the results. Reformulate to research questions.

Response: The section now reads (line 81ff): “Here, we report on a series of experiments, asking whether Microbembex monodonta wasps locate their hidden burrow entrances relative to small local landmarks and whether the view to the wider panorama is needed for pinpointing the burrow. By studying the wasps’ search when their nest entrances were covered in a variety of ways we also tested whether local visual, olfactory and auditory/vibrational cues help wasps to locate their hidden nests.” 

L90: it’s actually quite important that longitude is West (so -89.579446). I suggest to write it as one of the following: (46.172763N, 89.579446W), (46.172763, -89.579446)

Done.

L104: ‘trained’: what does this actually refer to? If it is was is described in the previous graph, indicate there what the training phase is. Suggestion (if I am correct in interpreting what the training is): L100 rewrite as ‘Each wasp was allowed to return undisturbed, enter, and leave. This is the training phase of the wasp to the objects.’

Response: Done. We added (line 103ff); “This is the phase in which the wasp is trained to recognize the objects.”

L114: Refer to fig 2 here (if I’m right).

Response: No. Fig. S2 is appropriate.

L124: make sure to split the sentences with commas if a part is some form of a dependent clause. It’s of course partially a personal taste, but I like it when the writer guides me in where to ‘pause’ when reading (you have to assume as a writer that readers are lazy). The pauses are often obvious to a writer, but this is not always so for a reader. I tried to indicate it in the comments where I think extra comma’s or hyphens (or rather em dashes) would have eased my reading: take these as suggestions. For instance, here, I would a comma between ‘foraging’ and ‘a triangular’. Please also keep this in mind when re-reading your manuscript to add these where confusion with clauses might be possible.

Done

L127: I wonder if the wasp was not confused already then by the experiment, because you removed already the local cues she probably used to replace them by your own?

Response: The sentence now reads (line 131): “The wasp was allowed to get used to this disturbance and to return, enter, exit, and cover her burrow.”

L132: add something to explain this results in 6cm to the left of the burrow.

Response: We are not sure what the referee means here.

Influence of non-visual cues, experiment 1: this part already improved a lot in clarity compared to the previous version. I have some additional comments/suggestions to add to the clarity:

- Already refer in the beginning of this section to table 1

- State in the beginning what the aim is (explore different types of possible cues?)

- State what your expectations are a priori for each of the different materials. This is somewhat deducible from table 1, but deserves more explicit, elaborate explanation in the text (maybe somewhere around L140?). This would also make it more clear, step by step, to the reader how the experiment tests the different types of cues.

Response: The section now reads (line 141ff): “Experiment 1: This experiment was designed to mask or alter local homing cues within the burrow vicinity - visual, olfactory, tactile, and possibly auditory/vibratory (Table 1). The distribution of repeated digging attempts of returning wasps was recorded with the expectation that the absence of crucial cues would lead to no or more widely distributed digging attempts. Seven different covers were placed over the burrows of 12 wasps, each designed to mask or alter at least one potential homing cue that might help the wasps to precisely locate their burrow entrances. Each cover is referred to as condition and the name of each condition begins with the part closest to the land surface, followed in upward placement order of the subsequent parts. The conditions are shown in Fig. 1.”

- L150 and further: as in the explanation for ‘condition (A)’, state which cues are excluded/included for each condition. It is summarized in table 1, but some written out explanation in the text and captions should complement this. 

- L159 and further: as previous, state the hypothesis or what is excluded/included and elaborate. It is summarized in table 1, but in the text you can then give more explanation to for instance, the question marks in the table.

Response: We now add this information for each condition.

- L163: plexiglass should be in italics

Done 

- L168: what is exactly the purpose of this hole?

Response: The hole allowed the wasps access to potential olfactory and auditory/vibrational cues. We point this out now in line 186ff: “This would enable any olfactory and auditory/vibrational cues to emanate, unlike the sealed window in condition (C)). This condition thus masked other visual and sand relief cues, but left olfactory and auditory/vibrational cues unmodified.”

- L160, 161: 256 or 49 3 cm squares are hard to imagine, I would prefer 16 by 16 (or 16×16) squares with sides of 3 cm. Same for 21 cm² on L161; L164: give exact dimensions of fiberboard frame; give diameter of what is cut from the centre (hole or square?); L174: 256 3cm squares: give the dimensions as well.

Done

- L160: 11.0 is also in inches? And could you rescale those to cmxcm?

Done

- Table1: 1) the uncertain states deserve more explanation, as in a previous comment, explain more in the text. 2) for E: cloth alone hugging surface: why is olfactory ruled out for this one? Deduced from the other conditions, I thought this was the one making the olfactory possible. And why would the cloth be masking the auditory cue?

Response: We realize that we probably had been too cautious about the effects of different covers on potential cues and have simplified Table 1. We now clarify the expected effects on potential cues by the different conditions in the method section line 168ff, added caveats where we cannot be absolutely certain about effects and have changed Table 1 accordingly. We also clarified expected effects in Fig. 5. Basically, Conditions A & B have a plexiglass cover which would seriously interfere with auditory/vibrational cues; D would allow both olfactory and auditory/vibrational cues to emanate from the hidden nest entrance and in conditions C & D the area was covered with paper, there would thus be no visual cue from the sand relief.

L176: omit ‘to me’

Done

L182: omit ‘for me’

Done

L203: repeat the objective here (as it was not explicitly stated for experiment 1). But I would not say it is similar to it, but rather an alternative to see if only visual cues are enough to pinpoint the nest.

Response: We now state the objectives at the beginning of the Experiment 1 section.

L206: give exact dimensions instead of squared centimeters.

Done

General for results: state at the beginning of each part shortly what was tested again (readers are lazy), either by explaining it in the first sentence or adding a shot title (rather than “experiment 1”). For instance, on L229: ‘Figure 3B show the responses of one of the wasps to visual barrier that blocks distant visual cues.’

Response: We now have added explanatory sentences to each of the experiments in the results section.

L233: ‘some’ is quite vague, can you quantify? [now line254]

Response: Sentence deleted.

L253: explain a bit more in the material and methods (with LXX) that you moved the barriers. Here, in the results section, it comes out of nowhere.

Response: Method section now reads (line 117) “The barriers were at least 60 cm high, but differed in configuration for each of the wasps and could be moved sideways to cover or uncover different parts of the panorama.”

L285: omit ‘of the’ (ellipses)

Response: We reworded (line 314ff): “Greater homing accuracy is shown by smaller ellipses centred on the burrow entrance location.”

L346-348: I don’t really understand the azimuth/translation explanation. What does this mean for the orientation of the wasp, does it first set the angle correct and then approaches the goal? Can you elaborate on this?

Response: We now clarify (line 380ff): “ that minimized first the azimuth mismatch between remembered and currently seen objects by changes in orientation and subsequently the apparent size mismatch by moving toward objects that appear smaller than remembered and away from objects that appear larger than remembered.”

L349-352: is this then the generalized model (as mentioned in the introduction; ‘global image matching’)? Does this one then expand on the snapshot model (as mentioned in the introduction), or is it a contrasting/competing model? Some extra explanation what the difference between both is could be helpful here. Then also end this paragraph with stating that your results support the latter model more. Rewriting the part in the introduction and discussion regarding snapshot and global image matching might be needed to synchronize both parts.

Response: The difference is captured in the two sentences at the end of this paragraph: the original snapshot model assumed that the visual scene can be segmented into discrete individual objects. This is difficult in complex natural scenes, but may not be needed, because global image differences provide the same information. These are not mutually exclusive models: they both operate on visual information that is in principle available to the insects.

L361: comma between ‘rings’ and ‘found’

Done

L365: em dashes, or commas, after Tinbergen and between ‘triangulum’ and ‘found’.

Done

L369: what is ‘motion parallax’? When introducing a new term, elaborate on what it means and why it is relevant for your study.

Response: We avoid using the term motion parallax, but rather explain what it means directly (line 403ff)“: as a homing insect moves closer to the goal, the images of close objects move faster across its visual field than those of more distant objects”

L393: ‘to home more easily/rapidly’?

Response: changed to “more efficiently”

L414: a dot to much at the end of the sentence.

Done

L452: ‘in front’ instead of forward (I was misguided by the ‘forward’ term as I first interpreted this as an actual overshoot).

Done

L455: I doubt you can talk here about ‘significantly’ different from A, as the ‘statistical’ comparison is only with A (see major comments); if the spread of B is indeed higher than others, my first idea would be to think about the multitude of material layers added to this condition (or the frame, the spread seems high for the conditions with the frame, eyeballing the ellipses in Fig. 5).

Response: Changed to “possible”.

L467, 469, 475: replace utilize by use

Done

L469: put ‘:’ after ‘as such’, because the next sentence explains the ‘no functional distinction’. 

Done

I would also nuance this statement more, because you did not test homing to the broad burrow vicinity, but only that in the burrow vicinity, a panorama is crucial for locating the nest entrance (which is a very interesting thing in itself). So I would not make a conclusion stating that homing to the nest aggregate is functionally similar to local nest recognition, as the former has not been tested. It probably uses landscape features as well on a larger scale, but this has not been tested. I suggest to omit this statement, and as a replacement stress the importance of the surrounding landscape.

Response: We modified this sentence (line 504ff): “There may thus be no functional distinction between homing to the broad burrow vicinity (although this remains to be investigated) and recognition of the burrow location as such:…”

---

## [Decision Letter · Decision Letter 2]

5 Jan 2023

PONE-D-21-38862R2DIGGER WASPS MICROBEMBEX MONODONTA SAY (HYMENOPTERA, CRABRONIDAE) RELY EXCLUSIVELY ON VISUAL CUES WHEN PINPOINTING THEIR NEST ENTRANCESPLOS ONE

Dear Dr. Zeil,

Thank you for submitting your manuscript to PLOS ONE. After careful consideration, we feel that it has merit but does not fully meet PLOS ONE’s publication criteria as it currently stands. Therefore, we invite you to submit a revised version of the manuscript that addresses the points raised during the review process.

I have read the manuscript myself and had the same problems as the reviewer – I am not convinced by the notched boxplots for a number of reasons, some of which are trivial, others not.

1) For starters, the way the figure is set up will lead to the botchplots being very small on a page – it will be difficult to properly assess overlap

2) as the reviewer pointed out, dependency of the data points in the different groups (repeated measures) are nothing notched boxplots take into account. I would actually think that taking this into account will make the differences even more clear

3) notched boxplots are not made for multiple comparisons. In theory you compare each of the boxes with each of the others, which would require some serious adjustment of p-values. We just can’t interpret notches like that. I do think that the way you set this all up is the best possible solution for these kinds of problems – you actually compare all treatments to a single (no-cue) control. However, in combination with the low sample sizes I do not think the presentation of the results purely based on the notched boxplots is appropriate and I would urge you to revise it.

There are multiple possible ways to improve the presentation of the data, for example:

- a repeated measures anova or a Friedman test would at least give us information on whether there are any differences between the treatments. Based on this I would bemuch more inclined to trust the box plots

-I would actually favour plotting individual data points with such a sample size

the summary output of a linear model in R would actually test each treatment against a single control – which would be fitting if that control was the treatment that did not offer any cues.

We look forward to receiving your revised manuscript.

Kind regards,

Volker Nehring

Academic Editor

PLOS ONE

Journal Requirements:

Additional Editor Comments:

I had some further comments, please see below. Importantly, I often had problems to understand what the actual data points were

line 43 “to precisely locate their burrows. The shifting of the latter causes homing wasps”

That sounds like you mean digging within the burrows

67topography or even the sound of moving larvae emanating from the nest may help ground-nesting

68insects to pinpoint the exact location of the entrance.

Any reference for the larval sounds? If not, please make it more obvious that this is your own speculation

137 5) The influence of non-

This should be 4), shouldn’t it?

Fig 3a: I don’t understand what its shown. According to the text these figures are the attempts of a “typical wasp” – but only five wasps were tested and 6 wasps are shown? I also don’t understand the explanations for f, g and h. The figure caption implies that the wasps always dug where the cork configuration would indicate – but it seems b, c , d and h did not dig in the centre of the triangle? Was the triangle not centered around the burrow? Where were the burrows previously? It can’t be the black dot because in c the dot does not line up with the x either. There is really too little information for me to understand what is shown and what the data points are.

Similar problem in Fig 3B -the tests says these are the responses of one of the wasps, but in the Figure it seems like these are responses of different wasps.

Lines 267-272 Please be more specific here – how many wasps were tested with which of the types of barriers?

Fig 4: I don’t find it clear what is shown on the x-axis. what does “responding to” mean? Is this the number of wasps searching where the artificial landmarks indicate, or where the burrow really was? I would also suggest to add statistics to Figure 4C (e.g. a chi-squared test).

To be honest, after seeing the data in Fig. 4 I wonder if Fig 3 is necessary at all.

304seven conditions described in the method section and in each of the panels in Fig. 5. Each condition

305graph shows the areas for 99% of all attempted diggings as grey ellipses and the means of the search

How many digging attempts per wasp?

Fig 5 what’s the definition of an outlier here?

Lines 361-428: A lot of this text feels like a repition or expancsion of what was already said in the introduction. I would find it more appropriate in that section

Dataset S3: Since these are repeated measures it wouold be good if you could add the IDs of the individuals to each line of the data set. In general I it was unclear how many digging attempts were observed per wasp. Perhaps it would be best to also include the coordinates of each digging attempt.

Reviewers' comments:

Reviewer's Responses to Questions

**Comments to the Author**

1. If the authors have adequately addressed your comments raised in a previous round of review and you feel that this manuscript is now acceptable for publication, you may indicate that here to bypass the “Comments to the Author” section, enter your conflict of interest statement in the “Confidential to Editor” section, and submit your "Accept" recommendation.

Reviewer #1: (No Response)

2. Is the manuscript technically sound, and do the data support the conclusions?

Reviewer #1: Yes

3. Has the statistical analysis been performed appropriately and rigorously? 

Reviewer #1: No

4. Have the authors made all data underlying the findings in their manuscript fully available?

Reviewer #1: Yes

5. Is the manuscript presented in an intelligible fashion and written in standard English?

Reviewer #1: Yes

6. Review Comments to the Author

Reviewer #1: The authors provide a series of interesting experiments to study the scale and type of cues used for nest recognition in the digger wasp Microbembex monodonta. I really look forward to using this manuscript for education and as a reference in my own work regarding nest recognition behavior.

However, there is in my opinion still one fundamental issue with the framing of the statistics used. Notch boxplots are indeed an accepted piece of descriptive statistics, but that does not mean it is applicable to any kind of experimental design, as in this case one with repeated measurements. I give suggestions below how to frame the statistics used more descriptive and qualitative. I also added some last (very) minor comments. This work is almost ready to be published, but the fundamental statistical concept I point to is too crucial to be ignored (if you don’t use the appropriate statistics for the experimental design at hand, your significance values are unreliable).

Note: the line numbers I refer to are to the non-track-change revised manuscript.

Major comment on the use of notch boxplots

Firstly, yes they are indeed an accepted piece of descriptive statistics, but it does not mean it is very well suited to every experimental design. As explained in my previous review, there were repeated measurements involved, but the sample size does not allow to properly take this into account. You can indeed still use such visualizations descriptively, but cannot conclude anything regarding significance as the statistics are strictly speaking not applicable for the experimental design. I would thus suggest to omit any reference to ‘significant differences’ and use the notch boxplots as purely qualitative/descriptive. At the following lines such ‘significance’ is mentioned: L308-309, L319-321, L466, L476. Please omit these and be more descriptive about it (such as ‘indications for’, ‘seems to be’, ‘there is a visual difference in distribution’, etc…). Then as we cannot talk about ‘significant’ difference with this experimental design/statistics, omit the part in the discussion on the second difference (Condition A, compared with Condition B, D) L477-479, as you mentioned in your response that the difference is only 1 cm. Of course, in the results you can still add that there seems to be a small, probably negligible, difference for that case. But it’s not worth mentioning in the discussion when there is nothing to discuss about.

Secondly, what you described in your responses, that the main insight is that these distributions are essentially very similar, was an eye-opener for me that this should be the main focus of the results and not the few small, probably negligible, differences. It is a very important interpretation that should be integrated in the text itself, I would suggest in the results (in paragraph L314-322) and in the discussion on L466.

Thus, in conclusion, shift the focus in both results and discussion to the fact that these distributions are essentially very similar, but there does seem to be a difference with condition E and F which has a reasonable explanation, which you discuss. Omit any notion of significance in both results as discussion as you can only apply descriptive statistics and discuss the distributions qualitatively.

Minor comments:

L22: Hymenoptera with capital

L67, L448-453: as you mentioned in your previous response, this is a hypothesis of yourself and very suggestive. State it as such if you don’t have any reference to back-up such a statement. But maybe there is some literature out there that points to sounds of larvae of insects being a cue to conspecifics (beetles, social wasps?)?

L71: ‘…is a solitary…’

L118: ‘check’ instead of ‘ask’ (conducting an experiment is the step after asking a certain research question).

L164, 170: 21.6cmx27.9cm

L172: insert enter before condition C

L216: omit one dot (two dots present)

L218: ‘…similar to the…’

L221: 5.7cmx5.7cm

L268: space between ‘C,D’

L301: excluding instead of exclude

L305-313: this is explanation about a method you used, so move this to the results section. Then the results are immediately about what you see in the figure. The explanation in the caption of the figure itself can stay.

L364: add closing bracket ‘)’

L383, 410, 501: the ‘van’ Iersel and ‘van den’ Assem are actually part of the surname (they are Dutch names), so it should actually read (also in the reference): ‘van Iersel and van den Assem [10]….’

L413: ‘A’ should not be a capital; 30cmx50cm

L503: please add references to this final statement (what are the recent experimental and modelling evidence you talk about?).

7. PLOS authors have the option to publish the peer review history of their article (what does this mean?). If published, this will include your full peer review and any attached files.

Reviewer #1: **Yes: **Femke Batsleer

---

## [Author Response · Author response to Decision Letter 2]

28 Jan 2023

I am grateful to both Volker and Femke for their diligent reading of, and their constructive comments on the manuscript. I hope to have comprehensively addressed them as indicated below, with the biggest change being a statistical analysis of the results presented in Fig. 5. Line numbers refer to the TrackChange version of the manuscript.

Sadly, my co-author Matt Cormons, who did the experiments presented in this manuscript has passed away in early December last year.

Comments by Volker Nehring, Academic Editor:

I have read the manuscript myself and had the same problems as the reviewer – I am not convinced by the notched boxplots for a number of reasons, some of which are trivial, others not.

1) For starters, the way the figure is set up will lead to the botchplots being very small on a page – it will be difficult to properly assess overlap

2) as the reviewer pointed out, dependency of the data points in the different groups (repeated measures) are nothing notched boxplots take into account. I would actually think that taking this into account will make the differences even more clear

3) notched boxplots are not made for multiple comparisons. In theory you compare each of the boxes with each of the others, which would require some serious adjustment of p-values. We just can’t interpret notches like that. I do think that the way you set this all up is the best possible solution for these kinds of problems – you actually compare all treatments to a single (no-cue) control. However, in combination with the low sample sizes I do not think the presentation of the results purely based on the notched boxplots is appropriate and I would urge you to revise it.

There are multiple possible ways to improve the presentation of the data, for example:

- a repeated measures anova or a Friedman test would at least give us information on whether there are any differences between the treatments. Based on this I would bemuch more inclined to trust the box plots

-I would actually favour plotting individual data points with such a sample size

the summary output of a linear model in R would actually test each treatment against a single control – which would be fitting if that control was the treatment that did not offer any cues.

Response: It is important to clarify that in the experiments shown in Fig. 5, the information on wasp IDs and the location of individual digging attempts has been lost, but that otherwise the data set is balanced, because each wasp contributes the same number of data (mean and spread of 15 digging attempts) to each condition and each of 12 wasps was confronted with each condition. This has now been clarified in line 210ff.

Linear model results are now tested with ANOVA and in addition with a permutation test using the F-statistics of the linear model as the permutation statistics, both showing that there are no significant differences in how wasps respond to the different conditions. A t-test was employed to test differences between the x- and y-directions of means and spread across all conditions, which are significant. 

Statistics are now described in line 214ff and results are presented in line 354ff.

In addition, individual data points are now shown for each conditions, together with their median. See Fig. 5H and legend line 378ff.

Additional Editor Comments:

I had some further comments, please see below. Importantly, I often had problems to understand what the actual data points were

line 43 “to precisely locate their burrows. The shifting of the latter causes homing wasps”. That sounds like you mean digging within the burrows

Response: The sentence now reads (line 43ff): ‘The shifting of such local objects causes homing wasps to dig for their burrows relative to the shift…’

67topography or even the sound of moving larvae emanating from the nest may help ground-nesting insects to pinpoint the exact location of the entrance. Any reference for the larval sounds? If not, please make it more obvious that this is your own speculation

Response: Now referring to (line 67): Müller A, Obrist MK (2021) Simultaneous percussion by the larvae of a stem-nesting solitary bee – a collaborative defence strategy against parasitoid wasps? Journal of Hymenoptera Research 81: 143–164. https://doi.org/10.3897/jhr.81.61067

137 5) The influence of non-This should be 4), shouldn’t it?

Response: Corrected.

Fig 3a: I don’t understand what its shown. According to the text these figures are the attempts of a “typical wasp” – but only five wasps were tested and 6 wasps are shown? I also don’t understand the explanations for f, g and h. The figure caption implies that the wasps always dug where the cork configuration would indicate – but it seems b, c , d and h did not dig in the centre of the triangle? Was the triangle not centered around the burrow? Where were the burrows previously? It can’t be the black dot because in c the dot does not line up with the x either. There is really too little information for me to understand what is shown and what the data points are.

Response: Figure 3 has now been simplified and a better description is offered both in the legend and in the text:

The figure legend now clarifies (line 269ff): ‘Fig. 3 - The influence of local and distant cues. (A) A typical example of how one of 5 wasps responded to shifts of local landmarks. Three cork stoppers (gray circles) were introduced around the burrow entrance (black dots) of a wasp. (a) training trial for the wasp to learn introduced cork-configuration cue surrounding the burrow prior to first shift. (b) – (h) consecutive configuration-shifts and digging attempts (⊗) of this particular wasp, which eventually does find her burrow; each initial digging following a subsequent shift was where the wasp was last successful, relative to the configuration. At (g) the wasp was not permitted to find her burrow before the next shift, resulting in her digging relative to configuration in (h) at a location where she had found the burrow in (f). (B) A wasp responding to the presence of an opaque barrier behind the burrow, with digging attempts at either end of the barrier with unimpeded view of the wider surrounding. (C) Same wasp digging at the ends of the barrier when the latter is centered on the burrow, then digging at either end, when the barrier is shifted to the left and right of the burrow.’

The text now refers to the simplified figure (line 249ff): ‘Local landmarks influence where wasps search for their hidden burrow entrances (Fig. 3A): Five wasps were allowed to become accustomed to finding their burrow in the center of an array of 3 cork stoppers (‘training’) and on subsequent returns were confronted with a shifted array of these landmarks. In each shift-trial wasps eventually located their burrow. All responded similarly to the one shown in Fig. 3A. Before attempting to dig for their burrows the wasps reacted by flying around the shifted configuration of local landmarks for two to three minutes, almost always where the configuration led them and their initial digging attempt was always at a location where, relative to the landmark array, they had found their burrow on their previous return (see [56]) for more evidence of this rapid location learning). Eventually the wasps corrected themselves and dug at their burrow entrances, which had not been disturbed; they did so by first hovering over the place they had been misled to dig, then flew directly above the actual burrow entrance, landed, and dug.’

Similar problem in Fig 3B -the tests says these are the responses of one of the wasps, but in the Figure it seems like these are responses of different wasps.

Response: Figure 3B has been removed, including related section in the legend and in the method section (line 104ff).

Lines 267-272 Please be more specific here – how many wasps were tested with which of the types of barriers?

Response: line 298 now clarifies that 6 wasps were tested, but only one made digging attempts as indicated in Fig. 3B and C.

Fig 4: I don’t find it clear what is shown on the x-axis. what does “responding to” mean? Is this the number of wasps searching where the artificial landmarks indicate, or where the burrow really was?

Response: Now clarified in the text (line 305ff): ‘To investigate how local and distant cues interact when wasps attempt to locate their hidden burrow entrances, their response to small shifts of local landmarks was tested in the presence of a large opaque barrier. Both digging attempts at the center of a three-cube landmark array and hovering above it were considered as an indication that the wasps used the visual appearance of the landmarks to locate their burrow. The barrier set-up for these experiments is shown in Fig. 4A and the sequence of tests with local visual cues in Fig. 4B. Results for all 10 wasps are shown in Fig. 4C. All 10 wasps learnt to use the cube-configuration as a homing cue after a single exposure (Fig. 4B, a). When tested with the shifted cube-configuration (Fig. 4B, b) 8 wasps first dug at the center of the shifted array before locating their burrow. The two that did not follow landed and dug without hesitation at their actual burrows. One of the eight wasps that located their burrow did not do so immediately; she hovered over the point where the cubes misled her. However, she quickly shifted to her actual burrow location, landed, and successfully dug.’

And line 330ff: ‘Once the barrier was in place, masking frontal, and some lateral, distant cues all 10 wasps failed to dig at the center of the landmark array, none hovered above it and none did find her burrow (Fig. 4C, c). The wasps flew back and forth in front of the barrier, around, and over it (see Fig. 3B), and even flew among the cubes. They never attempted to dig, never hovered over any particular area, nor otherwise indicated any recognition of the cube-configuration; they appeared to be lost. Yet, all 10 wasps had been able to enter the vicinities of their burrows. When the barrier was removed nine of the 10 wasps pinpointed the nest in the center of the landmark array without difficulty; the tenth wasp did not return.’

And in the figure legend (line 325ff): ‘(C) The number of wasps responding to the shift of the landmark configuration by digging attempts or by hovering at the center of the landmark array in condition (b), (c), and (d) (see (B) above). None of the wasps followed the configuration in (c).’

I would also suggest to add statistics to Figure 4C (e.g. a chi-squared test).

Response: I am advised that one cannot do a chi-squared test with zero responses. I do think, however, that the results are very clear without statistics.

To be honest, after seeing the data in Fig. 4 I wonder if Fig 3 is necessary at all.

Response: The barrier/local landmark experiment shown in Fig. 4 demonstrates that local landmarks can only be used by the insects as a visual aid to burrow location, if they are seen in the context of the full landmark panorama. This is additional information, compared to what has been shown in Fig. 3, namely (a) that local landmarks are attended to, (b) that that screens affect the wasps’ ability to locate the burrow and (c) that they tend to dig at (inappropriate) locations where at least the distant landmark panorama matches their burrow location memory.

304seven conditions described in the method section and in each of the panels in Fig. 5. Each condition graph shows the areas for 99% of all attempted diggings as grey ellipses and the means of the search. How many digging attempts per wasp?

Response: Each wasp was allowed 15 digging attempts for each condition. This is now clarified in lines 205, 212, line 342 & and in legend l368

Fig 5 what’s the definition of an outlier here?

Response: Bar graphs are now only shown for all data across all conditions, while the rest of the panels show individual data for each condition. An outlier is a value that is more than 1.5 times the interquartile range away from the bottom or top of the box. Now clarified in legend line 376ff.

Lines 361-428: A lot of this text feels like a repition or expancsion of what was already said in the introduction. I would find it more appropriate in that section

Response: I’d like to keep this section because it allowed us to discuss the details of present results (such as the influence of local and distant cues) in the context of previous work. This would be difficult to do in the introduction without stressing the patience of readers.

Dataset S3: Since these are repeated measures it wouold be good if you could add the IDs of the individuals to each line of the data set. In general I it was unclear how many digging attempts were observed per wasp. Perhaps it would be best to also include the coordinates of each digging attempt.

Response: The ID of wasps and the coordinates of their individual digging attempt have unfortunately been lost. This is now clarified in method section line 210ff.

Comments by Femke Batsleer:

Reviewer #1: The authors provide a series of interesting experiments to study the scale and type of cues used for nest recognition in the digger wasp Microbembex monodonta. I really look forward to using this manuscript for education and as a reference in my own work regarding nest recognition behavior.

However, there is in my opinion still one fundamental issue with the framing of the statistics used. Notch boxplots are indeed an accepted piece of descriptive statistics, but that does not mean it is applicable to any kind of experimental design, as in this case one with repeated measurements. I give suggestions below how to frame the statistics used more descriptive and qualitative. I also added some last (very) minor comments. This work is almost ready to be published, but the fundamental statistical concept I point to is too crucial to be ignored (if you don’t use the appropriate statistics for the experimental design at hand, your significance values are unreliable).

Response: We now present statistics (line 214ff), but it is important to note that information on wasp ID and individual digging attempts have been lost (line 210ff). Please see my responses to similar concerns by Volker Nehring above.

Major comment on the use of notch boxplots

Firstly, yes they are indeed an accepted piece of descriptive statistics, but it does not mean it is very well suited to every experimental design. As explained in my previous review, there were repeated measurements involved, but the sample size does not allow to properly take this into account. You can indeed still use such visualizations descriptively, but cannot conclude anything regarding significance as the statistics are strictly speaking not applicable for the experimental design. I would thus suggest to omit any reference to ‘significant differences’ and use the notch boxplots as purely qualitative/descriptive. At the following lines such ‘significance’ is mentioned: L308-309, L319-321, L466, L476. Please omit these and be more descriptive about it (such as ‘indications for’, ‘seems to be’, ‘there is a visual difference in distribution’, etc…). Then as we cannot talk about ‘significant’ difference with this experimental design/statistics, omit the part in the discussion on the second difference (Condition A, compared with Condition B, D) L477-479, as you mentioned in your response that the difference is only 1 cm. Of course, in the results you can still add that there seems to be a small, probably negligible, difference for that case. But it’s not worth mentioning in the discussion when there is nothing to discuss about.

Response: We now do present a statistical analysis (linear models, permutation test and t-test) that shows that the null-hypothesis, namely that the responses of wasps to the different conditions all come from the same distribution cannot be rejected (line 353ff). Which makes any comparison of responses to individual conditions superfluous (line 522ff). 

Secondly, what you described in your responses, that the main insight is that these distributions are essentially very similar, was an eye-opener for me that this should be the main focus of the results and not the few small, probably negligible, differences. It is a very important interpretation that should be integrated in the text itself, I would suggest in the results (in paragraph L314-322) and in the discussion on L466.

Thus, in conclusion, shift the focus in both results and discussion to the fact that these distributions are essentially very similar, but there does seem to be a difference with condition E and F which has a reasonable explanation, which you discuss. Omit any notion of significance in both results as discussion as you can only apply descriptive statistics and discuss the distributions qualitatively.

Response: See response above and text line 522ff.

Minor comments:

L22: Hymenoptera with capital

Response: Corrected.

L67, L448-453: as you mentioned in your previous response, this is a hypothesis of yourself and very suggestive. State it as such if you don’t have any reference to back-up such a statement. But maybe there is some literature out there that points to sounds of larvae of insects being a cue to conspecifics (beetles, social wasps?)?

Response: Reference added ([53] Müller A, Obrist MK (2021) Simultaneous percussion by the larvae of a stem-nesting solitary bee – a collaborative defence strategy against parasitoid wasps? Journal of Hymenoptera Research 81: 143–164. doi.org/10.3897/jhr.81.61067

L71: ‘…is a solitary…’

Response: Corrected.

L118: ‘check’ instead of ‘ask’ (conducting an experiment is the step after asking a certain research question).

Response: Corrected.

L164, 170: 21.6cmx27.9cm

Response: Corrected.

L172: insert enter before condition C

Response: Corrected.

L216: omit one dot (two dots present)

Response: Corrected.

L218: ‘…similar to the…’

Response: Corrected.

L221: 5.7cmx5.7cm

Response: Corrected.

L268: space between ‘C,D’

Response: Corrected.

L301: excluding instead of exclude

Response: Corrected.

L305-313: this is explanation about a method you used, so move this to the results section. Then the results are immediately about what you see in the figure. The explanation in the caption of the figure itself can stay.

Response: You probably meant moving this to the Method section. I’d prefer to leave it here, so that readers do not need to go back to the method section.

L364: add closing bracket ‘)’

Response: Corrected.

L383, 410, 501: the ‘van’ Iersel and ‘van den’ Assem are actually part of the surname (they are Dutch names), so it should actually read (also in the reference): ‘van Iersel and van den Assem [10]….’

Response: Corrected.

L413: ‘A’ should not be a capital; 30cmx50cm

Response: Corrected.

L503: please add references to this final statement (what are the recent experimental and modelling evidence you talk about?).

Response: Review added [65] Zeil J. Visual navigation: properties, acquisition and use of views. Journal of Comparative Physiology A 2022. doi.org/10.1007/s00359-022-01599-2

---

## [Editor Report · Decision Letter 3]

9 Feb 2023

Digger wasps Microbembex monodonta SAY (Hymenoptera, Crabronidae) rely exclusively on visual cues when pinpointing their nest entrances

PONE-D-21-38862R3

Dear Dr. Zeil,

We’re pleased to inform you that your manuscript has been judged scientifically suitable for publication and will be formally accepted for publication once it meets all outstanding technical requirements.

Kind regards,

Volker Nehring

Academic Editor

PLOS ONE

Additional Editor Comments (optional):

I am sad to learn of the passing of Matt Cormons and I am sorry that the overall review process took so long.

I like the revised version and found the experiments very easy to follow now. I think all reviewer comments were adequately addressed. I have two further comments for consideration, but how to deal with them is up to the author:

1) From the metadata in the submission system it appears as if this might end up being a single author paper. The manuscript contains language indicating multiple authors, which might be confusing (“one of us observed..”).

2) If it is still available it would be nice if the exact locations of the digging attempts would be published as part of S3 (which summarizes the 15 individual data points per individual). Perhaps these data might be useful to someone in the future.
---

## [Editor Report · Acceptance letter]

14 Feb 2023

PONE-D-21-38862R3 

Digger wasps *Microbembex monodonta* SAY (Hymenoptera, Crabronidae) rely exclusively on visual cues when pinpointing their nest entrances 

Dear Dr. Zeil:

I'm pleased to inform you that your manuscript has been deemed suitable for publication in PLOS ONE. Congratulations! Your manuscript is now with our production department. 

Kind regards, 

on behalf of

Dr. Volker Nehring 

Academic Editor

PLOS ONE